# TINNs: Time-Induced Neural Networks for Solving Time-Dependent PDEs

**Chen-Yang Dai** [1]  **Che-Chia Chang** [2]  **Te-Sheng Lin** [1 3]  **Ming-Chih Lai** [1]  **Chieh-Hsin Lai** [1]

## Abstract

Physics-informed neural networks (PINNs) solve time-dependent partial differential equations (PDEs) by learning a mesh-free, differentiable solution that can be evaluated anywhere in space and time. However, standard space–time PINNs take time as an input but reuse a single network with shared weights across all times, forcing the same features to represent markedly different dynamics. This coupling degrades error performance and can destabilize training when enforcing PDE, boundary, and initial constraints jointly. We propose *Time-Induced Neural Networks (TINNs)*, a novel architecture that parameterizes the network weights as a learned function of time, allowing the effective spatial representation to evolve over time while maintaining shared structure. The resulting formulation naturally yields a nonlinear least-squares problem, which we optimize efficiently using a Levenberg–Marquardt method. Experiments on various time-dependent PDEs show up to $4\times$ improved relative $L^2$ error and $10\times$ faster convergence compared to PINNs and strong baselines. Code is available at https://github.com/CYDai-ml/TINN.

## 1. Introduction

Time-dependent partial differential equations (PDEs) are central to modeling transport, fluid flows, and reaction–diffusion phenomena. While classical numerical solvers based on discretization (e.g., finite differences and finite elements (LeVeque, 2007; Brenner & Scott, 2008)) are highly accurate, they can become costly when the domain geometry is complex, when solutions must be queried at arbitrary locations, or when the effective dimension grows due to parameters, uncertainty, or inverse settings.

Physics-informed neural networks (PINNs) provide an appealing alternative by learning a surrogate solution that satisfies the governing equations through a constrained training objective (Raissi et al., 2019). By minimizing a weighted combination of PDE residuals and initial/boundary constraints at collocation points, PINNs yield a mesh-free, differentiable solution operator that can be queried anywhere in the domain (Stiasny et al., 2021; Cuomo et al., 2022; Luo et al., 2025).

Despite these successes, standard space–time PINNs still struggle to accurately model dynamics whose spatial complexity evolves over time. In the prevailing formulation, time is treated as an additional input coordinate and a single neural network $u_{\boldsymbol{\theta}}(\mathbf{x}, t)$ is trained over the entire space–time domain with a shared set of parameters $\boldsymbol{\theta}$. As a result, time influences the model only through input conditioning, while the deeper representation is reused across all temporal regimes. We refer to this structural limitation as the *time-entanglement problem*: qualitatively different regimes, such as smooth early-time behavior and sharp later-time transitions, must be represented by the same shared features. This causes representation interference and makes optimization difficult when jointly enforcing the PDE residual, boundary conditions, and initial conditions.

To build intuition, consider a toy parameterization in one spatial dimension with a single affine space–time feature,

$$u_{\boldsymbol{\theta}}(x, t) := U(wx + vt + b),$$

where $U$ represents the remaining (possibly deep) nonlinear network, and $\boldsymbol{\theta} = (w, v, b)$ consists of three learnable scalars. Its spatial derivative is

$$\partial_x u_{\boldsymbol{\theta}}(x, t) = U'(wx + vt + b)\, w.$$

This exposes the bottleneck: time affects the network only through an additive shift $vt$, while the spatial scaling $w$ is fixed for all $t$. Therefore, the model cannot directly increase spatial steepness over time by rescaling spatial features.

[1]Department of Applied Mathematics, National Yang Ming Chiao Tung University, Hsinchu 30010, Taiwan [2]Institute of Artificial Intelligence Innovation, National Yang Ming Chiao Tung University, Hsinchu 30010, Taiwan [3]National Center for Theoretical Sciences, National Taiwan University, Taipei 10617, Taiwan. Correspondence to: Chen-Yang Dai <cydai.sc12@nycu.edu.tw>, Te-Sheng Lin <teshenglin@nycu.edu.tw>, Chieh-Hsin Lai <chiehhsinlai@gmail.com >.

*Proceedings of the 43rd International Conference on Machine Learning*, Seoul, South Korea. PMLR 306, 2026. Copyright 2026 by the author(s).

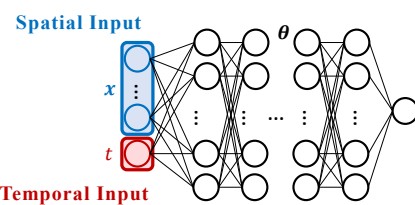

**Vanilla Neural Network $u_{\boldsymbol{\theta}}(\boldsymbol{x}, t)$**

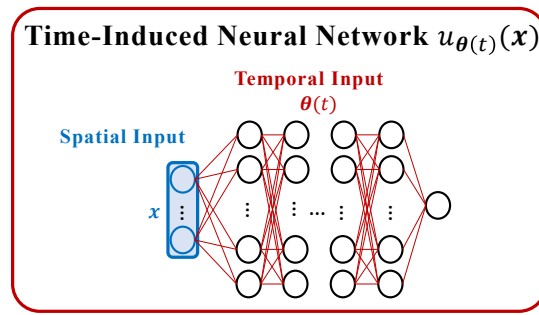

**Time-Induced Neural Network $u_{\boldsymbol{\theta}(t)}(\boldsymbol{x})$**

*Figure 1.* **Space–time PINN (time as input) vs. TINN (time in parameter space).** The left shows how a vanilla neural network structure deals with time-dependent PDEs. The time is incorporated into the input dimension. On the right-hand side, the time information in TINN is integrated into the parameter space, making it easier to capture the complex dynamics of the system.

Instead, any steepening must be achieved indirectly by shifting the argument of $U'(\cdot)$. For example, at $x = 0$ the slope changes from $U'(b)w$ at $t = 0$ to $U'(v + b)w$ at $t = 1$. This shift-only control is brittle in practice, since large changes in spatial gradients must be produced by moving across activation regimes. Deeper networks may increase expressivity, but the same coupling persists: time still modulates the solution through shared features, so evolving spatial scales remain implicit rather than explicitly controlled.

Existing methods address these issues mainly via optimization (Wang et al., 2023; Bihlo, 2024; Wang et al., 2025) and training heuristics, including adaptive sampling, loss reweighting, and causality-inspired curricula that prioritize earlier times (McClenny & Braga-Neto, 2023; Wang et al., 2024a;b; Bischof & Kraus, 2025). While often effective, they retain the global form $u_{\boldsymbol{\theta}}(\mathbf{x}, t)$, forcing one shared representation across all times and handling temporal non-stationarity only through input conditioning.

To address the *time-entanglement problem*, we propose *Time-Induced Neural Networks (TINNs)*, which represent a time-dependent solution as a trajectory in *parameter space*:

$$u_{\boldsymbol{\theta}(t)}(\mathbf{x}),$$

with smoothly varying parameters $\boldsymbol{\theta}(t)$. Unlike standard space–time PINNs that fit all time with a single parameter vector, TINNs learn a mapping $t \mapsto \boldsymbol{\theta}(t)$, producing a family of time-indexed spatial networks (see in Figure 1). This explicit temporal evolution allows spatial features and gradients to adapt over time, reducing representation interference and enabling accurate solutions with compact models.

Empirically, TINNs achieve strong error performance with compact models without uniformly increasing capacity.

Building on the flexibility of TINNs, we exploit an efficient training procedure tailored to the resulting objective. Since the loss naturally takes a nonlinear least-squares form (from the PDE residual together with boundary and initial con-

straints), we adopt a Gauss–Newton–style optimizer based on the Levenberg–Marquardt (LM) method (LEVENBERG, 1944; Marquardt, 1963). In practice, this second-order update improves stability by better balancing competing constraints than first-order methods.

Our contributions are: (i) we propose *Time-Induced Neural Networks (TINNs)*, a time-induced parameterization with practical constructions of $\boldsymbol{\theta}(t)$ to deal with *time-entanglement problem* in space–time PINNs; (ii) we employed an LM-based optimizer that exploits the resulting nonlinear least-squares structure; and (iii) we validate TINNs on benchmark time-dependent PDEs, showing improved accuracy and stability over strong baselines.

The rest of the paper is organized as: Section 2 reviews PINNs and related work. Section 3 introduces TINNs and our parameterization of $\boldsymbol{\theta}(t)$. Section 4 presents experiments, and Section 5 discusses mechanisms. Section 6 concludes.

## 2. Preliminary and Related Work

### 2.1. Physics-Informed Neural Networks (PINNs)

Consider a time-dependent PDE posed on a spatial domain $\Omega \subset \mathbb{R}^d$ and a time interval $[0, T]$:

$$\begin{cases} \mathcal{L}(u) = 0, & \text{in } \Omega \times [0, T], \\ \mathcal{B}(u) = 0, & \text{on } \partial\Omega \times [0, T], \\ \mathcal{I}(u) = 0, & \text{on } \Omega \times \{0\}, \end{cases}$$

where $\mathcal{L}$ is a differential operator (possibly nonlinear), and $\mathcal{B}$ and $\mathcal{I}$ denote the boundary and initial condition operators, respectively. Physics-informed neural networks (Raissi et al., 2019) approximate the solution by a neural network $u_{\boldsymbol{\theta}}(\mathbf{x}, t)$ and enforce these constraints by minimizing a weighted sum

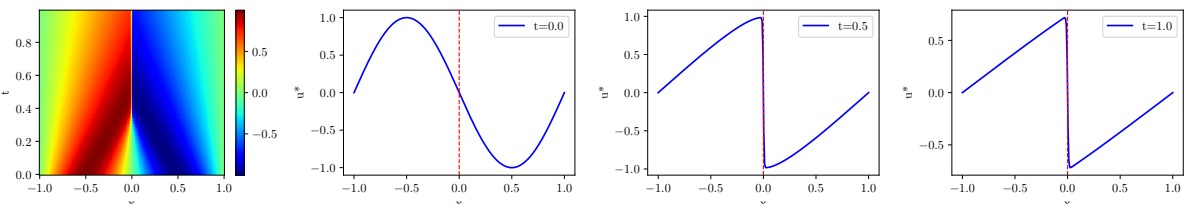

*Figure 2.* **Motivation for TINNs on viscous Burgers. Left:** space–time contour of the exact solution on $x \in [-1, 1]$, $t \in [0, 1]$. **Right:** solution profiles at $t \in \{0, 0.5, 1.0\}$. The solution evolves from a smooth profile to a thin transition layer near $x = 0$, producing much steeper spatial gradients at later times. Although the profiles at $t = 0.5$ and $t = 1.0$ are similar in shape, their sharpness differs substantially, which is challenging for a single space–time network with shared parameters and motivates TINN's time-modulated parameterizations.

of squared residuals evaluated at collocation points:

$$L(\boldsymbol{\theta}) := \frac{\lambda_r}{N_r} \sum_{i=1}^{N_r} \left\| \mathcal{L}(u_{\boldsymbol{\theta}})(\mathbf{x}_i^r, t_i^r) \right\|_2^2$$

$$+ \frac{\lambda_b}{N_b} \sum_{i=1}^{N_b} \left\| \mathcal{B}(u_{\boldsymbol{\theta}})(\mathbf{x}_i^b, t_i^b) \right\|_2^2 + \frac{\lambda_{ic}}{N_{ic}} \sum_{i=1}^{N_{ic}} \left\| \mathcal{I}(u_{\boldsymbol{\theta}})(\mathbf{x}_i^{ic}, 0) \right\|_2^2,$$

$$\tag{1}$$

where $\lambda_r, \lambda_b, \lambda_{ic} \geq 0$ are penalty weights for the residual, boundary condition, and initial condition terms of the PDE, respectively. The collocation sets $\{(\mathbf{x}_i^r, t_i^r)\}_{i=1}^{N_r} \subset \Omega \times [0, T]$, $\{(\mathbf{x}_i^b, t_i^b)\}_{i=1}^{N_b} \subset \partial\Omega \times [0, T]$, and $\{(\mathbf{x}_i^{ic}, 0)\}_{i=1}^{N_{ic}} \subset \Omega \times \{0\}$ correspond to the constraint of the residual, boundary, and initial condition of the PDE, respectively.

For time-dependent problems, standard space–time PINNs treat $t$ as an additional input dimension and parameterize the solution with a single time-independent parameter vector $\boldsymbol{\theta}$ shared across all time steps, i.e., a network $u_{\boldsymbol{\theta}} : \mathbb{R}^{d+1} \to \mathbb{R}$, trained by minimizing Equation (1) over the entire space–time domain.

## 2.2. Related Work

For time-dependent PDEs, several PINN variants have been developed to improve training stability over long time horizons. Causal PINNs (Wang et al., 2024b) address error propagation by applying a time-adaptive loss reweighting that prioritizes early-time error performance, since inaccuracies near small $t$ can accumulate and degrade the solution at later times. Despite this improvement in optimization, the method still relies on a single space–time network that treats time as an additional input coordinate, which can limit representational efficiency for complex temporal dynamics.

Another direction explicitly separates space and time (Datar et al., 2026) by using a fixed set of spatial features (often randomly initialized) and learning only time-varying coefficients on top of them. The temporal evolution is then advanced with a classical ODE solver, which can be fast and accurate; however, this hybrid design departs from

fully end-to-end PINN training and typically requires additional mechanisms to enforce initial and boundary conditions. Moreover, the overall error is often dominated by the accumulation of numerical errors introduced by the underlying time integration scheme. For example, we applied both the separation method and TINN to a transport equation. TINN achieves a relative $L^2$ error of $8.42 \times 10^{-10}$, whereas the separation method attained $3.04 \times 10^{-8}$, demonstrating higher accuracy within a fully neural, end-to-end framework. The detail of the experiment is in Appendix E.1.

In contrast, we adopt a space–time separation viewpoint while retaining continuous, physics-informed, end-to-end learning. We parameterize the solution with a time-dependent neural representation and optimize it directly using the standard PINN objective, without freezing spatial features or relying on discrete time integration. Furthermore, prior work identifies challenges (Krishnapriyan et al., 2021; Rathore et al., 2024), focusing on **why** PINNs are difficult to train. In our work, TINN addresses **how** temporal information is incorporated. This leads to improved trainability and is orthogonal to existing strategies (Wang et al., 2024b; Wu et al., 2024), enabling straightforward combination.

## 3. Time-Induced Neural Networks (TINNs)

### 3.1. Limitation with Vanilla Network Parametrizations

As discussed in the introduction, standard space–time PINNs suffer from the *time-entanglement problem*: temporal evolution is entangled with spatial features via additive shifts in activation arguments, making evolving spatial scales difficult to represent explicitly. Here, we illustrate this limitation on a concrete time-dependent PDE, and show how it can be addressed by our proposed TINN, which introduces time-dependent network parameters.

To illustrate the missing degree of freedom, we consider a toy TINN in one spatial dimension with a single affine space–time feature,

$$u_{\boldsymbol{\theta}(t)}(x) := U\big(w(t)x + b(t)\big),$$

where $U$ denotes the remaining network and the parameters $\boldsymbol{\theta}(t)$ vary smoothly with time. Its spatial derivative is

$$\partial_x u_{\boldsymbol{\theta}(t)}(x) = w(t)\, U'\big(w(t)x + b(t)\big).$$

Unlike vanilla space–time networks, temporal evolution can now modify spatial steepness directly through the time-dependent scaling $w(t)$, rather than indirectly through shifts inside $U'(\cdot)$. This explicit control of spatial scales is precisely what is absent in standard PINN parameterizations.

The time-entanglement problem becomes evident, for instance, in viscous Burgers' equation (defined in Appendix B): the solution evolves from a smooth profile into a thin yet continuous transition layer. As shown in Figure 2, the overall shape remains similar while spatial gradients increase markedly over time. Since time affects the representation only through input conditioning, standard space–time PINNs can express this steepening only implicitly via temporal shifts, which is difficult in practice. The mathematical details explaining why TINN is effective when solution gradients sharpen over time are provided in Appendix A.1.

We additionally report the absolute error of the spatial derivative at $x = 0$ over $t \in [0, 1]$ in Figure 3. Since no closed-form derivative is available, we compute a high-resolution reference solution of the viscous Burgers equation using `Chebfun` package (Driscoll et al., 2014) and obtain its derivative via spectral differentiation. For both the vanilla multilayer perceptron (MLP) and TINN with comparable model sizes, we compute derivatives via automatic differentiation.

As shown in Figure 3, the derivative error for the MLP increases sharply after the shock-like transition forms, reflecting its difficulty in resolving increasingly steep spatial gradients at later times. In contrast, TINN maintains a substantially smaller and more stable error throughout the entire time interval. This behavior highlights a structural limitation of standard space–time PINNs: with a single shared representation across time, they struggle to adapt to evolving spatial scales. Allowing spatial features to evolve explicitly over time, as in TINN, alleviates this issue.

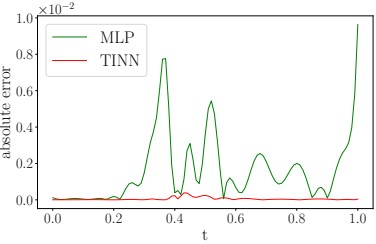

*Figure 3.* **Absolute error of the spatial derivative at $x = 0$ over $t \in [0, 1]$: vanilla MLP vs. TINN.** With comparable parameter counts, TINN yields smaller and more stable errors. See details in Appendix F.1.

Indeed, the time-entanglement problem can also arise in modern space–time parameterizations beyond MLPs (He

et al., 2016; Wang et al., 2021; Cho et al., 2023); we defer the discussion to Appendix C.1.

### 3.2. Designing $\boldsymbol{\theta}(t)$ for Efficient Training

To separate temporal evolution from spatial representation, we remove time from the network input and instead model the solution trajectory through time-dependent parameters. Specifically, we parameterize a time-dependent solution as

$$u(\mathbf{x}, t) := u_{\boldsymbol{\theta}(t)}(\mathbf{x}), \tag{2}$$

where $u_{\boldsymbol{\theta}} : \Omega \subset \mathbb{R}^d \to \mathbb{R}$ is a spatial neural network and $\boldsymbol{\theta}(t)$ varies smoothly over time. In this view, the backbone $u_{\boldsymbol{\theta}}$ captures spatial structure, while $\boldsymbol{\theta}(t)$ encodes the temporal dynamics as a trajectory in parameter space.

**Why Naive Hypernetworks of $\boldsymbol{\theta}(t)$ Are Expensive?** A key challenge in Equation (2) is to construct $\boldsymbol{\theta}(t)$ without introducing a prohibitive number of additional parameters. To make this concrete, consider an $L$-layer MLP,

$$u_{\boldsymbol{\theta}(t)}(\mathbf{x}) = \mathbf{W}_L\sigma\Big(\mathbf{W}_{L-1}\sigma\big(\cdots\sigma(\mathbf{W}_1\mathbf{x}+\mathbf{b}_1))+\mathbf{b}_{L-1}\Big)+\mathbf{b}_L,$$

parameterized by $\boldsymbol{\theta}(t) = \{(\mathbf{W}_\ell, \mathbf{b}_\ell)\}_{\ell=1}^L$, where $\sigma(\cdot)$ is a component-wise applied nonlinear activation, $\mathbf{W}_\ell \in \mathbb{R}^{l_\ell \times l_{\ell-1}}$ and $\mathbf{b}_\ell \in \mathbb{R}^{l_\ell}$ are time-dependent, with $l_0$ denoting the input (spatial) dimension. The total number of parameters is

$$N_D := \sum_{\ell=1}^L \big(l_{\ell-1}l_\ell + l_\ell\big).$$

A naive implementation of Equation (2) uses a fully-connected network to output $\boldsymbol{\theta}(t) \in \mathbb{R}^{N_D}$. If this time network has hidden width $h$, then its final layer alone requires $\mathcal{O}(N_D h)$ parameters. Since $N_D$ grows with the width of the spatial backbone (and typically increases with the PDE dimension through the input layer), this overhead can quickly dominate the overall model size and training cost.

This motivates restricting the temporal parameterization of $\boldsymbol{\theta}(t)$ to reduce dimensionality while retaining sufficient expressive power. A lightweight baseline is to impose a simple functional form on $\boldsymbol{\theta}(t)$. For example, one may assume a *linear trajectory* $\boldsymbol{\theta}(t) = \boldsymbol{w}t + \boldsymbol{b}$, or a *one-neuron trajectory* $\boldsymbol{\theta}(t) = \boldsymbol{w}_2\sigma(\boldsymbol{w}_1 t + \boldsymbol{b})$, where $\boldsymbol{w}, \boldsymbol{w}_1, \boldsymbol{w}_2, \boldsymbol{b} \in \mathbb{R}^{N_D}$. These choices substantially reduce the parameter count (roughly $2N_D$ and $3N_D$, respectively), but are often too restrictive to capture heterogeneous temporal behaviors across PDEs. As shown in Table 1, the linear trajectory works well for Allen–Cahn, while both linear and one-neuron behave similarly for Burgers, motivating a more flexible yet still compact design.

Taken together, these observations suggest two modeling requirements for the design of $\boldsymbol{\theta}(t)$. First, it should exhibit *macro-level coherence*: If the exact solution varies sharply

*Table 1.* **Simple parametric forms for $\theta(t)$ vs. TINN.** For fair comparison, we fix training points, iterations, optimizer settings, and random seeds, and match parameter budgets across methods. Linear and one-neuron trajectories are similar on Burgers, while linear is better on Allen–Cahn. Our proposed compact layer-wise construction of $\theta(t)$ (Equation (3)) achieves the best performance.

| Case | Rel $L^2$-Error ($\downarrow$) | # Params. | # Neurons |
|---|---|---|---|
| **Burgers** | | | |
| linear | 2.65E-06 | 1144 | 22 |
| one-neuron | 2.93E-06 | 1188 | 18 |
| TINN's $\theta(t)$ | **5.67E-07** | 1145 | 20 |
| **Allen–Cahn** | | | |
| linear | 3.25E-06 | 1188 | 22 |
| one-neuron | 1.47E-05 | 1242 | 18 |
| TINN's $\theta(t)$ | **2.73E-06** | 1185 | 20 |

around $t^\star$, then $\Phi(t)$ should be able to induce a correspondingly sharp, coordinated change in $\theta(t)$ near $t^\star$. Second, it should allow *micro-level diversity*: even near $t^\star$, layers should remain distinguishable and be allowed to evolve differently over time.

**Proposed Alternative: Compact Layer-wise Time Embedding.** To achieve both goals while avoiding a fully-connected network with an $N_D$-dimensional output, we propose a compact layer-wise code $\Phi(t) \in \mathbb{R}^{2L}$ ($2L-1$ when omitting the output bias), and lift it to the full parameter vector $\theta(t) \in \mathbb{R}^{N_D}$ via an entrywise affine map. This yields *macro-level coherence* (many parameters change in a coordinated manner when the solution varies sharply in time) while preserving *micro-level diversity* (different layers follow different temporal scalings). More precisely, we define the layer-wise embedding $\Phi(t)$ as:

$$\Phi(t) = (1 - \alpha)\, t \,+\, \alpha \odot \mathcal{N}(t),$$

where $\mathcal{N} : \mathbb{R} \to \mathbb{R}^{2L}$ is a small learnable network, $\alpha \in \mathbb{R}^{2L}$ is a learnable gate, $\odot$ denotes element-wise multiplication, and $1$ is the all-ones vector. We associate $\Phi_{2\ell-1}(t)$ and $\Phi_{2\ell}(t)$ with the $\ell$-th layer weight and bias groups $\mathbf{W}_\ell(t)$ and $\mathbf{b}_\ell(t)$, respectively. Given $\Phi(t)$, we construct the full parameter trajectory via a linear-affine lifting map $\mathbf{F}$ applied within each parameter group. Within a group, all entries share the same temporal coordinate of $\Phi(t)$, while retaining independent affine coefficients. For example, for $\mathbf{W}_1(t) = \{w_1^{ij}(t)\}_{i,j}$, we define

$$w_1^{ij}(t) = a_{w_1}^{ij}\, \Phi_1(t) + b_{w_1}^{ij},$$

and define the remaining weights and biases analogously. Equivalently, these entrywise affine rules together define a global mapping

$$\theta(t) \;=\; \mathbf{F}\big(\Phi(t)\big), \tag{3}$$

where $\mathbf{F}$ stacks all affine coefficients $\{a^{ij}, b^{ij}\}$ across layers and parameter entries. Thus, the learnable variables consist

of the embedding network $\mathcal{N}$, the gate $\alpha$, and the affine map $\mathbf{F}$. To avoid ambiguity, we denote by $\psi$ the collection of all trainable parameters in $\{\mathcal{N}, \alpha, \mathbf{F}\}$, and by $\theta(t)$ the induced time-dependent parameters of the spatial backbone. Once $\psi$ is trained, we can instantiate $\theta(t)$ (and hence $u_{\theta(t)}$) at any time $t$. Figure 4 illustrates the proposed construction. Our embedding is a compact parameterization designed to match the observed low-dimensional evolution of the fitted per-time networks, which also satisfies macro-level coherence and the micro-level diversity. A detailed discussion is provided in Appendix A.2.

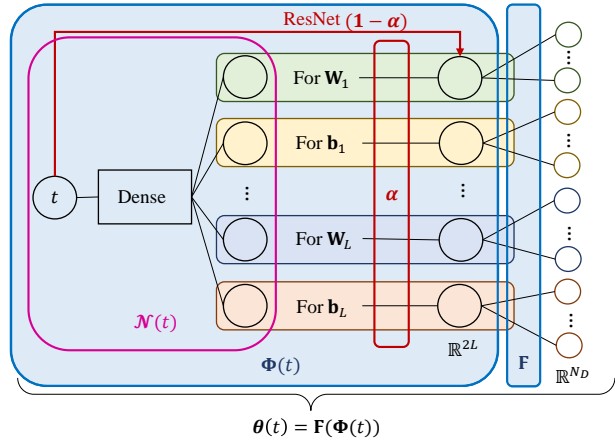

*Figure 4.* **Architecture for time-dependent parameters $\theta(t)$ in TINN.** A small dense network $\mathcal{N}(t)$ outputs a $2L$-dimensional code $\Phi(t)$, which is lifted to the full parameter vector $\theta(t) \in \mathbb{R}^{N_D}$ via an entrywise affine map, avoiding direct prediction of an $N_D$-dimensional output.

**Parameter Efficiency: Layer-wise Embedding vs. Naive Hypernetworks.** We now compare the parameter complexity of these two approaches. For a naive hypernetwork with hidden width $h$, the final layer alone requires $\mathcal{O}(N_D h)$ parameters to output $\theta(t) \in \mathbb{R}^{N_D}$.

In contrast, the TINN time network $\mathcal{N}(t)$ outputs the layer-wise code $\Phi(t) \in \mathbb{R}^{2L}$ and has $\mathcal{O}(Lh)$ parameters (for fixed depth). The subsequent entrywise affine lift $\theta(t) = \mathbf{F}(\Phi(t))$ introduces two coefficients per backbone entry, contributing $2N_D$ additional parameters. The total number of trainable parameters therefore scales as

$$\text{\# Params.} \;=\; 2N_D \,+\, \mathcal{O}(Lh),$$

which is substantially smaller than $\mathcal{O}(N_D h)$ in practice, since typically $N_D \gg L$ and $h > 2$. A concrete comparison is provided in Appendix F.1.

In this work, we use an MLP backbone, following common practice in the PINN literature to isolate the effect of the proposed temporal embedding structure. Nevertheless, our approach is not specific to MLPs and can be extended to other architectures; see Appendix C.2.

### 3.3. Fast TINN Training via Levenberg–Marquardt

Training PINNs naturally yields a *nonlinear least-squares* problem, in which the objective is a weighted sum of squared residuals of the governing PDE together with the initial and boundary conditions, evaluated at collocation points. TINN adopts the same physics-informed loss as standard space–time PINNs; the only difference is the parameterization $u_{\boldsymbol{\theta}(t)}(\mathbf{x})$, ensuring a fair comparison across methods.

Most PINN approaches rely on first-order optimizers such as Adam (Kingma & Ba, 2015) or quasi-Newton methods like L-BFGS (Liu & Nocedal, 1989), which do not explicitly exploit the least-squares structure and can be sensitive to scale mismatches among residual terms. While second-order methods are appealing in principle, naive Newton updates are often unstable due to poor conditioning.

We therefore adopt the Levenberg–Marquardt (LM) algorithm (LEVENBERG, 1944; Marquardt, 1963; Hu et al., 2024), a standard second-order method for nonlinear least-squares problems that interpolates between gradient descent and Gauss–Newton updates via adaptive damping. In our setting, LM operates on the physics-informed objective by forming and solving a sequence of damped linearized subproblems based on the stacked PDE, boundary, and initial residuals (see Appendix D for details).

Although LM can in principle be applied to general PINN parameterizations, its per-iteration cost scales with the number of trainable parameters, making it impractical for the large networks commonly used in recent PINN baselines (Duan et al., 2025; Wang et al., 2024a). By contrast, TINN is parameter-efficient yet expressive, keeping LM tractable for the moderately sized models considered here. Empirically, this combination yields faster and more stable convergence.

## 4. Numerical Results

### 4.1. Experimental Setup

To evaluate TINNs on complex temporal dynamics, we consider five time-dependent PDEs: Burgers, Allen–Cahn, Klein–Gordon, Korteweg–De Vries, and the wave equation (see Appendix B for details). All models are trained by minimizing the PINN objective in Equation (1).

**Implementation of TINNs.** To isolate the effect of the TINN parameterization, we use a unified training protocol across methods and avoid problem-specific heuristics. In particular, we do not employ random Fourier features (Tancik et al., 2020), constraint-preserving initializations (Wang et al., 2024a), causal/time-marching training (Wang et al., 2024b), or related techniques. This enables a direct and fair comparison to standard PINNs and baselines.

We instantiate TINN with a compact spatial backbone to improve parameter efficiency relative to standard space–time PINNs. The spatial network is a fully connected MLP with two hidden layers of width 20. Time dependence is introduced via an auxiliary network $\mathcal{N}(t)$, implemented as a two-hidden-layer MLP with width 10, which outputs the layer-wise embedding used to modulate backbone parameters. Both networks use tanh activations, and we omit the bias in the final output layer. The output dimension of $\mathcal{N}(t)$ matches the number of parameter groups in the backbone, which is five in our setup (weights and biases of the two hidden layers, and the output-layer weights).

The backbone input dimension matches the spatial domain dimension. For periodic boundary conditions (Allen–Cahn and Korteweg–De Vries), we additionally apply periodic input embeddings, which add two neurons to the second layer of the spatial network. Additional implementation details are provided in Appendix F.3.

**Baselines.** We compare TINNs against several representative baselines. For each baseline, we follow the best-performing training protocals. In addition, to isolate the effect of the proposed architectural components, we provide ablation studies under a unified training strategy in Appendix E.3.

*PINNs* (Raissi et al., 2019) serve as the vanilla baseline and are trained with Adam (Kingma & Ba, 2015) using a decaying learning-rate schedule.

*CoPINN* (Duan et al., 2025) extends SPINN (Cho et al., 2023) by dynamically reweighting samples based on estimated difficulty. While SPINN uses explicit space–time factorization, TINN employs implicit layer-wise temporal modulation via a hypernetwork, enabling more flexible representations. We denote our implementation as *CoPINN\**, as the released code is not optimized for best performance; we add learning-rate decay and high-precision matrix multiplication to further improve results.

*PirateNet* (Wang et al., 2024a) combines deep residual architectures with training enhancements (e.g., random Fourier features, causal training, and PDE-informed initialization). Following prior work, we train PirateNet using the improved second-order optimizer SOAP (Wang et al., 2025).

**Measurements.** We evaluate the relative $L^2$-error, training runtime, network size, and the error performance improvement (IMP). IMP measures how many times it is improved in relative $L^2$-error achieved by TINNs over other non-TINN baselines (see Appendix F.2).

### 4.2. Comparison with State-of-the-Art Baselines

The main results are summarized in Table 2. For a fair comparison, we implement all methods in JAX and evaluate

*Table 2.* **Results on various time-dependent PDEs.** All experiments run on a single NVIDIA A6000 GPU. We report relative $L^2$-error (mean over 5 runs), training time, and trainable parameters. **Bold** and underlined denote the best and second-best errors, respectively.

| Case | Rel $L^2$-Error ($\downarrow$) | Time ($\downarrow$) | # Params. | acc. IMP |
|---|---|---|---|---|
| **Burgers** | | | | |
| PINN (Raissi et al., 2019) | 2.19E-04 $\pm$ 1.65E-04 | 1.24hr | 309440 | 318$\times$ |
| CoPINN* (Duan et al., 2025) | 6.23E-05 $\pm$ 2.74E-05 | 0.78hr | 336864 | 90$\times$ |
| PirateNet SOAP (Wang et al., 2025) | 1.97E-06 $\pm$ 5.13E-07 | 1.70hr | 534853 | 2.9$\times$ |
| TINN (ours) | **6.89E-07 $\pm$ 3.97E-07** | **0.75hr** | 1145 | – |
| **Allen-Cahn** | | | | |
| PINN (Raissi et al., 2019) | 4.65E-01 $\pm$ 3.62E-01 | 0.95hr | 309760 | 1.21E+05$\times$ |
| CoPINN* (Duan et al., 2025) | 1.29E-04 $\pm$ 4.33E-05 | 0.80hr | 337464 | 33$\times$ |
| PirateNet SOAP (Wang et al., 2025) | 8.32E-06 $\pm$ 3.00E-06 | 1.50hr | 534981 | 2.2$\times$ |
| TINN (ours) | **3.85E-06 $\pm$ 1.48E-06** | **0.78hr** | 1185 | – |
| **Klein-Gordon** | | | | |
| PINN (Raissi et al., 2019) | 4.04E-03 $\pm$ 7.75E-03 | 1.53hr | 309760 | 845$\times$ |
| CoPINN* (Duan et al., 2025) | 6.61E-06 $\pm$ 4.84E-06 | 0.70hr | 212832 | 1.4$\times$ |
| PirateNet SOAP (Wang et al., 2025) | 1.88E-05 $\pm$ 6.11E-07 | 3.31hr | 379281 | 3.9$\times$ |
| TINN (ours) | **4.78E-06 $\pm$ 2.63E-06** | **0.67hr** | 1185 | – |
| **Korteweg-De Vries** | | | | |
| PINN (Raissi et al., 2019) | 2.47E-02 $\pm$ 9.93E-03 | 1.91hr | 309760 | 161$\times$ |
| CoPINN* (Duan et al., 2025) | 1.05E-01 $\pm$ 5.45E-02 | 1.14hr | 337464 | 686$\times$ |
| PirateNet SOAP (Wang et al., 2025) | 4.26E-04 $\pm$ 5.29E-05 | 1.86hr | 534981 | 2.8$\times$ |
| TINN (ours) | **1.53E-04 $\pm$ 4.73E-05** | **0.69hr** | 1185 | – |
| **Wave** | | | | |
| PINN (Raissi et al., 2019) | 5.01E-02 $\pm$ 1.89E-02 | 1.14hr | 309440 | 7.47E+03$\times$ |
| CoPINN* (Duan et al., 2025) | 3.89E-03 $\pm$ 1.67E-03 | 0.78hr | 336864 | 580$\times$ |
| PirateNet SOAP (Wang et al., 2025) | 2.88E-05 $\pm$ 8.89E-06 | 1.89hr | 534853 | 4.3$\times$ |
| TINN (ours) | **6.71E-06 $\pm$ 7.65E-06** | **0.77hr** | 1145 | – |

them under a *matched wall-clock training budget*, set to the runtime of TINNs. For each method, we report the final model obtained within this time budget. Methods with lower per-iteration cost therefore execute more optimization steps under the same budget; this includes PINN and CoPINN*, the latter benefiting from a more efficient automatic differentiation structure. For PirateNet, we follow the original experimental protocol and use the iteration count reported in the corresponding work, as its training procedure is not directly comparable under a wall-clock–matched setting.

Across all tested PDEs, TINNs achieve the best error performance while using substantially fewer trainable parameters than competing methods. For instance, TINNs improve relative $L^2$ error by 2.9$\times$ on Burgers and 2.2$\times$ on Allen–Cahn. Moreover, when trained with LM, TINNs converge markedly faster than the strongest baseline, PirateNet optimized with SOAP. On the Burgers equation, TINNs reach comparable or better error performance with a 10.55$\times$ speedup over PirateNet with SOAP, and achieve a 2.30$\times$ speedup on Allen–Cahn (more details in Appendix F.3 and Table 14).

Moreover, the number of unknowns in TINN does not grow significantly in high-dimensional settings, since the hyper-network output is independent of the spatial dimensionality. To further demonstrate this scalability, we apply TINN to the $(3 + 1)$–D Klein-Gordon equation and the $(3 + 1)$–D Navier-Stokes equation in Appendix B.8 and Appendix B.9, respectively. We also consider more challenging scenarios, including the chaotic Kuramoto-Sivashinsky equation, a $(2 + 1)$–D flow mixing problem, and the inverse problem for the Burgers equation; see Appendix B.6, Appendix B.7, and Appendix B.10, respectively.

Finally, although $\boldsymbol{\theta}(t)$ is parameterized by a neural network, computing higher-order temporal derivatives incurs no noticeable overhead in practice. In particular, evaluating $\partial_{tt}^2 u_{\boldsymbol{\theta}(t)}$ has comparable cost to $\partial_t u_{\boldsymbol{\theta}(t)}$: 30K training iterations require 0.75 hours for Burgers (using $\partial_t u_{\boldsymbol{\theta}(t)}$) and 0.77 hours for the wave equation (using $\partial_{tt}^2 u_{\boldsymbol{\theta}(t)}$).

### 4.3. Ablation Study

We further evaluate TINNs with Adam to decouple architectural gains from optimizer choice. While LM is most practical for small architectures, larger TINNs can be trained with Adam.

In Table 3, we train larger TINNs with Adam and compare against PirateNet under matched engineering choices.

*Table 3.* **Ablation of TINNs trained with Adam.** Results are averaged over five runs (seeds 0–4). TINNs achieve the best error performance across all cases, while using fewer parameters than PirateNet.

| Case | Rel $L^2$-Error | Time | # Params. |
|------|-----------------|------|-----------|
| **Burgers** | | | |
| PirateNet | 2.43E-05 | 1.16hr | 534853 |
| TINN | **2.15E-05** | 1.12hr | 283037 |
| **Allen-Cahn** | | | |
| PirateNet | 2.32E-03 | 0.99hr | 534981 |
| TINN | **8.29E-05** | 0.91hr | 283165 |
| **Klein-Gordon** | | | |
| PirateNet | 5.60E-05 | 2.90hr | 379281 |
| TINN | **4.98E-05** | 2.78hr | 107384 |

TINNs remain competitive on Burgers, Allen–Cahn, and Klein–Gordon, indicating that the gains stem from the time-induced parameterization and are not specific to LM. It also demonstrates that TINN with larger models can be trained for more complex PDEs.

*Table 4.* **Ablation of architectural choices under LM training.** We compare TINNs to subMLP, PirateNet, and a vanilla MLP with comparable parameter counts, all trained using LM. Results are averaged over five runs (seeds 0–4). TINNs achieve the best error performance

across all PDEs under similar model size and training time, highlighting the benefit of the time-induced parameterization.

| Case | Rel $L^2$-Error | Time | # Params. |
|------|-----------------|------|-----------|
| **Burgers** | | | |
| subMLP | 7.11E-05 | 0.75hr | 500 |
| MLP | 1.92E-05 | 0.82hr | 1160 |
| PirateNet | 9.17E-07 | 0.93hr | 1190 |
| TINN | **6.89E-07** | 0.75hr | 1145 |
| **Allen-Cahn** | | | |
| subMLP | 1.76E-03 | 0.80hr | 520 |
| MLP | 7.14E-06 | 0.79hr | 1202 |
| PirateNet | 4.09E-05 | 0.86hr | 1190 |
| TINN | **3.85E-06** | 0.78hr | 1185 |
| **Klein-Gordon** | | | |
| subMLP | 2.84E-05 | 0.68hr | 520 |
| MLP | 5.11E-06 | 0.68hr | 1202 |
| PirateNet | 8.16E-01 | 0.76hr | 1190 |
| TINN | **4.78E-06** | 0.67hr | 1185 |

Next, we conduct an LM-based ablation study over compact architecture choices; results are summarized in Table 4. We include a baseline, *subMLP*, which shares the same spatial backbone as TINNs but treats time $t$ as an additional input, yielding a standard space–time MLP. Since TINNs provide greater representational flexibility than subMLP, we expect improved performance under the same optimizer (see Section 5). In Table 4, *MLP* denotes a standard fully connected network, while *PirateNet* refers to a compact PirateNet architecture with a comparable number of parameters.

We also report the performance of CoPINN and PirateNet

under both single- and double- precision arithmetic in Appendix E.4. Furthermore, Appendix E.5 presents the training time and iteration counts required for TINN to achieve the error levels attained by other baselines. Finally, the performance of TINN under different optimizers is reported in Appendix E.6. Implementation details for the ablation studies are provided in Appendix F.4.

A related hypernetwork-based approach is *Hyper-PINN* (de Avila Belbute-Peres et al., 2021), which is designed for parameterized PDEs. In HyperPINN, the hypernetwork takes PDE parameters (e.g., the viscosity coefficient in the Burgers equation) as input and globally modulates the backbone by generating the entire set of network parameters, corresponding to the naive hypernetwork design discussed above. Empirically, TINN demonstrates consistently strong performance across all benchmarks, significantly outperforming HyperPINN on the Burgers and Klein–Gordon equations while remaining competitive on the Allen–Cahn equation, while also requiring less training time. Additional details are provided in Appendix E.2.

## 5. Mechanism of TINNs

**PINNs as a Restricted Case of TINNs.** Standard PINNs for time-dependent PDEs parameterize the solution with a single MLP $u_M(\mathbf{x}, t)$ taking $(d+1)$-dimensional inputs, where $d$ is the spatial dimension and time is treated as an additional coordinate. This can be viewed as a special case of our TINN framework, as formalized below.

**Proposition 5.1.** *Let $u_M(\mathbf{x}, t)$ be a MLP with $(d+1)$-dimensional input. Then $u_M$ can be viewed as a special case of a TINN in which time dependence appears only in the bias of the first layer.*

This can be interpreted as follows: for a fixed model size, the TINN function class strictly contains that of a vanilla space–time MLP; hence, TINNs are strictly more expressive. To see this, consider a standard space–time PINN parameterized by an $L$-layer MLP with input $(\mathbf{x}, t) \in \mathbb{R}^{d+1}$. Its first hidden layer takes the form

$$\mathbf{z} = \sigma\left(\mathbf{W}_1^x \mathbf{x} + \mathbf{W}_1^t t + \mathbf{b}_1\right),$$

where $\mathbf{W}_1^x \in \mathbb{R}^{l_1 \times d}$ is the spatial weight matrix, $\mathbf{W}_1^t \in \mathbb{R}^{l_1}$ is the time weight vector, and $\mathbf{b}_1 \in \mathbb{R}^{l_1}$ is the bias. The temporal contribution can be absorbed into a time-dependent bias, $\mathbf{b}_1(t) := \mathbf{b}_1 + \mathbf{W}_1^t t$, while all weight matrices remain time-independent and all subsequent layers receive no explicit temporal input.

In a TINN, each scalar parameter is parameterized in the form $a \mathbf{\Phi}_k(t) + b$. By setting $\mathbf{\Phi}_k(t) = t$ for the first-layer bias and choosing $a = 0$ for all weights and for all parameters in layers $\ell > 1$, the TINN reduces to a vanilla space–time PINN. Therefore, standard PINNs are a highly

restricted instance of TINNs, where temporal dependence is confined to the first-layer bias term.

**TINNs Provide Flexibility in Minimizing Equation (1).** In standard space–time PINNs, the PDE residual, boundary conditions, and initial conditions in Equation (1) are enforced simultaneously using a single network $u_{\boldsymbol{\theta}}(\mathbf{x}, t)$. As seen in the toy example, time enters only as an input, so all time slices share the same parameterization $\boldsymbol{\theta}$. This entangles temporal variation with spatial features, making time-varying spatial scales/gradients difficult to represent. We now further show that the same coupling also makes it challenging to satisfy the PDE residual and the initial/boundary constraints simultaneously.

To illustrate this limitation, consider the PDE

$$\begin{cases} \mathcal{L}u = 0, & (x, t) \in [-1, 1] \times [0, T], \\ u(x, 0) = u_{\text{ic}}(x), & x \in [-1, 1], \\ u(x, t) = u_{bc}(x, t), & (x, t) \in \{-1, 1\} \times [0, T], \end{cases}$$

where $\mathcal{L}$ is a differential operator. Suppose a vanilla neural network takes the form $U(\mathbf{W}_1^x x + \mathbf{W}_1^t t + \mathbf{b}_1)$, where $\mathbf{W}_1^x$, $\mathbf{W}_1^t$, $\mathbf{b}_1$ are the parameters of the first layer and $U$ represents the remaining network. Imposing the initial and boundary conditions requires

$$\begin{cases} U(\mathbf{W}_1^x x + \mathbf{b}_1) = u_{\text{ic}}(x), & x \in [-1, 1], \\ U(\mathbf{W}_1^x + \mathbf{W}_1^t t + \mathbf{b}_1) = u_{\text{bc}}(1, t), & t \in [0, T], \\ U(-\mathbf{W}_1^x + \mathbf{W}_1^t t + \mathbf{b}_1) = u_{\text{bc}}(-1, t), & t \in [0, T]. \end{cases}$$

Thus, the distinction between the initial condition and the boundary conditions (and between $x = 1$ and $x = -1$) is introduced only through the first-layer affine inputs $\mathbf{W}_1^x x + \mathbf{W}_1^t t + \mathbf{b}_1$. All subsequent layers share the same parameters and must jointly satisfy all constraints through a single representation, limiting the model's flexibility to accommodate different components of the PINN loss.

In contrast, a TINN represents the solution as $u_{\boldsymbol{\theta}(t)}(x)$ with time-dependent parameters $\boldsymbol{\theta}(t)$. The initial and boundary constraints become

$$u_{\boldsymbol{\theta}(0)}(x) = u_{\text{ic}}(x), \qquad u_{\boldsymbol{\theta}(t)}(\pm 1) = u_{\text{bc}}(\pm 1, t).$$

Unlike a space–time PINN, which must satisfy all constraints using a single shared parameterization, TINN allows *all* layers to adapt over time through $\boldsymbol{\theta}(t)$. This decoupling provides additional flexibility to fit the initial and boundary constraints while simultaneously reducing the PDE residual, leading to a more flexible optimization of Equation (1).

**TINNs Exhibit Substantially Reduced Overfitting.** On the viscous Burgers equation, whose solution develops a sharp shock-like transition, we monitor generalization using a validation loss evaluated on held-out collocation

points. While vanilla space–time PINNs exhibit a persistent train–validation gap and frequent loss spikes—even with collocation resampling—TINNs maintain a small gap, with validation loss tracking training loss closely throughout optimization (Figure 5). Correspondingly, TINNs show smoother loss decay and more stable parameter updates.

We attribute this behavior to a structural mismatch in standard space–time PINNs: a single network with shared parameters must fit distinct temporal regimes, which can cause representation interference and destabilize optimization. TINNs mitigate this by allowing the network parameters to evolve with time, better matching nonstationary dynamics and improving robustness. Although illustrated on Burgers' equation, the same issue is expected to arise broadly in time-dependent PDEs with strongly nonuniform temporal dynamics.

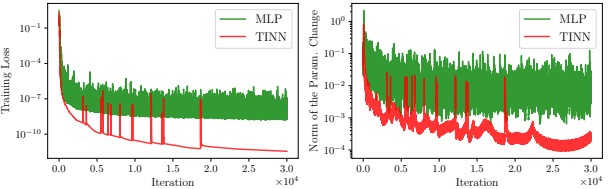

*Figure 5.* **Training stability on viscous Burgers: vanilla PINN vs. TINN. Left:** training loss. **Right:** $\ell_2$-norm of parameter updates. Despite resampling spikes, vanilla PINN shows high-variance loss and irregular parameter updates, whereas TINN converges more smoothly with more regular update magnitudes.

## 6. Conclusion

We identify a *time-entanglement* issue in standard space–time PINNs: a single network with shared parameters must fit heterogeneous temporal regimes, forcing time-varying spatial features into a fixed representation and often degrading optimization and generalization. We propose *Time-Induced Neural Networks (TINNs)*, which model temporal evolution as a smooth trajectory in parameter space, decoupling spatial structure from time variation. Across time-dependent PDE benchmarks, TINNs improve error performance and stability over strong PINN baselines while converging faster.

## Impact Statement

We propose *Time-Induced Neural Networks (TINNs)* for solving time-dependent PDEs. TINNs learn a time-indexed spatial networks to improve error performance and training stability, as an *AI for Science* approach. This paper aims to advance machine learning for scientific computing. We do not anticipate novel ethical concerns or direct negative societal impacts beyond those generally associated with machine learning research.

## Reproducibility Statement

We provide all the necessary code to reproduce TINN's main results on the benchmark PDEs (Burgers, Allen–Cahn, Klein–Gordon, Korteweg–de Vries, and Wave) in the accompanying supplementary zip file. Detailed experimental configurations are provided in Appendix F to support faithful reproducibility.

## Acknowledgements

T.-S. Lin acknowledge the supports by National Science and Technology Council, Taiwan, under research grants 111-2628-M-A49-008-MY4 and 114-2124-M-390-001. M.-C. Lai acknowledge the support by National Science and Technology Council, Taiwan, under research grant 113-2115-M-A49-014-MY3.

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

# Contents

## A. Mathematical Discussion

### A.1. TINN Helps when Gradient Sharpen over Time

Consider the one-hidden-layer space–time model (MLP)

$$u_M(x,t) = \sum_{j=1}^{m} c_j\, \sigma(w_j x + v_j t + b_j).$$

If $\sigma$ has bounded derivatives (e.g. $\sigma = \tanh$), then

$$\|\partial_x u_M(\cdot,t)\|_\infty \le \|\sigma'\|_\infty \sum_{j=1}^{m} |c_j w_j|, \qquad \|\partial_{xx} u_M(\cdot,t)\|_\infty \le \|\sigma''\|_\infty \sum_{j=1}^{m} |c_j|\, |w_j|^2,$$

and

$$\|\partial_{xt} u_M(\cdot,t)\|_\infty \le \|\sigma''\|_\infty \sum_{j=1}^{m} |c_j|\, |w_j v_j|,$$

These bounds are independent of $t$, so spatial and mixed derivatives are controlled by a single time-independent envelope.

When the target evolves from smooth to sharp (e.g. Burgers), a single parameter set must capture both regimes. Large $w_j$ enables steep gradients but introduces high-frequency artifacts at early times; small $w_j$ preserves smoothness but limits late-time resolution. This intrinsic trade-off reflects time-entanglement between spatial scale and temporal evolution.

For the corresponding TINN model

$$u_T(x,t) = \sum_{j=1}^{m} c_j(t)\, \sigma(w_j(t)x + b_j(t)),$$

the analogous bounds become

$$\|\partial_x u_T(\cdot, t)\|_\infty \leq \|\sigma'\|_\infty \sum_{j=1}^{m} |c_j(t)w_j(t)|, \qquad \|\partial_{xx} u_T(\cdot, t)\|_\infty \leq \|\sigma''\|_\infty \sum_{j=1}^{m} |c_j(t)|\, |w_j(t)|^2,$$

so the envelope varies with $t$. This enables adaptive spatial scaling over time and removes the above coupling.

## A.2. Layer-wise Time-Embedding Approach

We introduce a low-dimensional temporal code

$$\Phi(t) = (1 - \alpha)t + \alpha \odot N(t), \qquad \Phi(t) \in \mathbb{R}^{2L}$$

(or $\mathbb{R}^{2L-1}$ when the output bias is omitted), and lifts it to the full parameter vector through an entrywise affine map. For parameter group $i$ and entry $p$,

$$\theta_{i,p}(t) = a_{i,p}\Phi_i(t) + b_{i,p}.$$

*Concrete example.* For a one-hidden-layer network, this yields

$$u(x,t) = \sum_{j=1}^{m}(\Phi_3(t)c_j + \tilde{c}_j)\sigma\left((\Phi_1(t)w_j + \tilde{w}_j)x + (\Phi_2(t)b_j + \tilde{b}_j)\right).$$

Here, $\Phi_1(t)$ rescales the input weights, controlling spatial frequency, while $\Phi_3(t)$ modulates the output amplitude. If $\Phi_1(t)$ increases over time, the network progressively sharpens its spatial features.

This makes the two effects precise.

*Macro-level coherence.* All entries in the same group share the same scalar driver $\Phi_i(t)$. They may differ in magnitude and sign through $a_{i,p}$, but they follow the same temporal profile. More globally, all groups are driven by coordinates of the same low-dimensional code $\Phi(t)$. Thus the network evolves on a low-dimensional temporal manifold, rather than assigning an independent time function to each parameter.

*Micro-level diversity.* Despite the shared driver within a group, different entries still have different affine coefficients $a_{i,p}, b_{i,p}$. Hence they need not have the same scale, sign, or offset, and different groups may follow different coordinates $\Phi_i(t)$. The model therefore enforces shared temporal structure without forcing identical parameter motion.

This intuition can be formalized as a low-dimensional trajectory. For each group,

$$\bar{\theta}_i(t) = a_i\, \Phi_i(t) + b_i,$$

so its trajectory lies in a two-dimensional affine subspace, and after centering in a one-dimensional linear subspace. Thus, each group evolves along a single dominant temporal mode.

This is consistent with a diagnostic experiment on the KdV equation, where we fit an independent spatial network at each time step using a classical solver and observe that the resulting parameter trajectories are nearly rank-1: the leading singular component explains 93% to 99% of the variation for weights and 87% to 89% for biases (see in Table 5).

*Table 5.* Fraction of squared singular-value energy captured by the top singular value, i.e., $\sigma_1^2 / \sum_i \sigma_i^2$.

| Matrix | $\sigma_1^2 / \sum_i \sigma_i^2$ |
|---|---|
| $W_1(t)$ | 95% |
| $b_1(t)$ | 87% |
| $W_2(t)$ | 93% |
| $b_2(t)$ | 89% |
| $W_3(t)$ | 99% |

## B. Time-Dependent PDEs Problem

This section summarizes the partial differential equations covering shocks, phase transitions, nonlinear wave propagation, and oscillatory dynamics, used in our numerical experiments. Figure 6 shows spatiotemporal heatmaps of the reference solutions for the 1D PDE benchmarks (Burgers, Allen–Cahn, KdV, and Wave). We further apply TINN to more complicated dynamic system, high-dimensional PDEs, and inverse problem, which are also summarized in the section.

### B.1. Viscous Burgers Equation

We consider the one-dimensional viscous Burgers equation on the space–time domain $[-1, 1] \times [0, 1]$. The solution $u(x, t)$ satisfies

$$\begin{cases} u_t + u\,u_x - \nu u_{xx} = 0, & (x, t) \in (-1, 1) \times (0, 1), \\ u(x, 0) = -\sin(\pi x), & x \in [-1, 1], \\ u(-1, t) = u(1, t) = 0, & t \in [0, 1], \end{cases}$$

where the viscosity parameter is set to $\nu = 0.01/\pi$. We use the same reference solution as in (Raissi et al., 2019) for quantitative evaluation.

### B.2. Allen-Cahn Equation

We consider the one-dimensional Allen–Cahn equation on the space–time domain $[-1, 1] \times [0, 1]$. The solution $u(x, t)$ satisfies

$$\begin{cases} u_t - 0.0001 u_{xx} + 5u^3 - 5u = 0, & (x, t) \in (-1, 1) \times (0, 1), \\ u(x, 0) = x^2 \cos(3\pi x) + x^2, & x \in [-1, 1], \end{cases}$$

with periodic boundary conditions in space. The reference solution is generated using the Chebfun package (Driscoll et al., 2014) with 512 Fourier modes and a time-step size of $10^{-6}$.

In several prior works (Raissi et al., 2019; Wang et al., 2024a; 2025), the Allen–Cahn equation is initialized with $u(x, 0) = x^2 \cos(\pi x)$, which is incompatible with periodic boundary conditions. Although the initial value matches on the boundary, its spatial derivatives do not. This mismatch introduces an $L^\infty$ error of order $10^{-3}$ near the initial time, with the largest error occurring near $t = 0$. Consequently, all methods incur a relative $L^2$-error of order $10^{-5}$–$10^{-6}$ regardless of their approximation performance, resulting in an artificial error floor. In this regime, solution error comparisons among different methods become uninformative. To avoid this issue, we adopt the periodic-compatible initial condition above.

### B.3. Klein-Gordon Equation

We consider the two-dimensional Klein–Gordon equation on the space–time domain $[-1, 1] \times [-1, 1] \times [0, 10]$. The solution $u(x, y, t)$ satisfies

$$\begin{cases} u_{tt} - \Delta u + u^2 = f, & (x, y, t) \in (-1, 1) \times (-1, 1) \times (0, 10), \\ u(x, y, 0) = x + y, & (x, y) \in [-1, 1] \times [-1, 1], \\ u_t(x, y, 0) = 2xy, & (x, y) \in [-1, 1] \times [-1, 1], \\ u(x, y, t) = u_{\mathrm{bc}}(x, y, t), & (x, y, t) \in \partial([-1, 1] \times [-1, 1]) \times [0, 10], \end{cases}$$

where $f$ denotes a source term and $u_{\mathrm{bc}}$ specifies the boundary condition. The exact solution is given by

$$u(x, y, t) = (x + y)\cos(2t) + xy\sin(2t),$$

from which both the source term $f$ and the boundary data $u_{\mathrm{bc}}$ are derived.

### B.4. Korteweg-De Vries Equation

We consider the one-dimensional Korteweg–de Vries (KdV) equation on the space–time domain $[-1, 1] \times [0, 1]$. The solution $u(x, t)$ satisfies

$$\begin{cases} u_t + u\,u_x + 0.022^2 u_{xxx} = 0, & (x, t) \in (-1, 1) \times (0, 1), \\ u(x, 0) = \cos(\pi x), & x \in [-1, 1], \end{cases}$$

with periodic boundary conditions in space. We use the same reference solution as in (Wang et al., 2024a) for quantitative evaluation.

### B.5. Wave Equation

We consider the one-dimensional wave equation on the space–time domain $[0, 1] \times [0, 1]$. The solution $u(x, t)$ satisfies

$$\begin{cases} u_{tt} - 4u_{xx} = 0, & (x, t) \in (0, 1) \times (0, 1), \\ u(x, 0) = \sin(\pi x) + \frac{1}{2}\sin(4\pi x), & x \in [0, 1], \\ u_t(x, 0) = 0, & x \in [0, 1], \\ u(0, t) = u(1, t) = 0, & t \in [0, 1]. \end{cases}$$

The exact solution is given by

$$u(x, t) = \sin(\pi x)\cos(2\pi t) + \frac{1}{2}\sin(4\pi x)\cos(8\pi t).$$

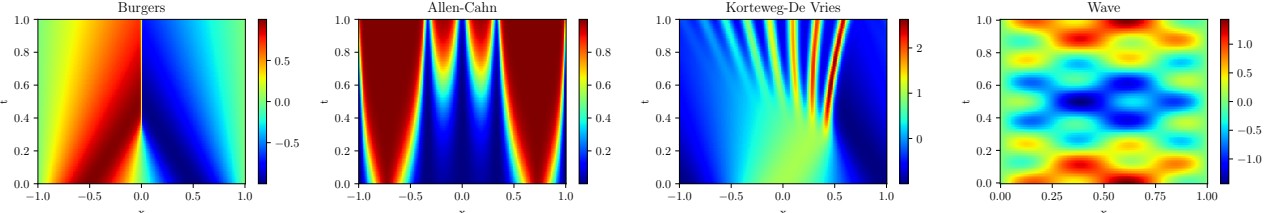

*Figure 6.* **PDEs solution.** The reference solutions for 1D PDEs are displayed.

### B.6. Kuramoto–Sivashinsky Equation

We consider the one-dimensional Kuramoto-Sivashinsky equation on the space-time domain $[0, 2\pi] \times [0, 0.8]$, which exhibits nonlinear instability and spatiotemporal chaos. The solution $u(x, t)$ satisfies

$$\begin{cases} u_t + \alpha u u_x + \beta u_{xx} + \gamma u_{xxxx} = 0, & (x, t) \in (0, 2\pi) \times (0, 0.8), \\ u(x, 0) = \cos(x)(1 + \sin(x)), & x \in [0, 2\pi], \end{cases}$$

with periodic boundary condition, where $\alpha = \frac{100}{16}$, $\beta = \frac{100}{16^2}$, $\gamma = \frac{100}{16^4}$. We apply TINN under ten time windows of 3329 parameters in each time window with the backbone $1 \times 20 \times 20 \times 9$, and the layer-wise time-embedding network, $2 \times 20 \times 20 \times 20 \times 20 \times 1$. Under the LM optimizer, we get the relative $L^2$ error $8.58E - 03$. The ground truth, the network prediction, and the error are shown in Figure 7.

### B.7. $(2 + 1)$-D Flow Mixing Problem

We consider the two-dimensional flow mixing problem on the space-time domain $[-4, 4] \times [-4, 4] \times [0, 4]$. The solution $u(x, y, t)$ satisfies

$$\begin{cases} u_t + au_x + bu_y = 0, & (x, y, t) \in (-4, 4) \times (-4, 4) \times (0, 4), \\ u(x, y, 0) = u_{ic}(x, y), & (x, y) \in [-4, 4] \times [-4, 4], \\ u(x, y, t) = u_{bc}(x, y, t), & (x, y, t) \in [-4, 4] \times [-4, 4] \times [0, 4], \end{cases}$$

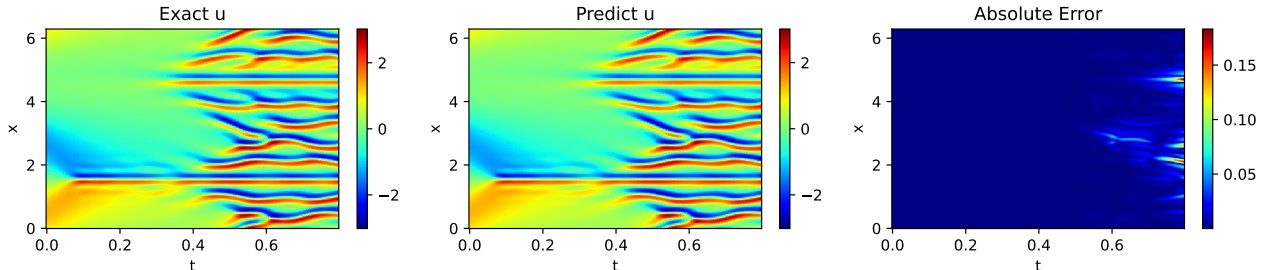

*Figure 7.* **Solution of the Kuramoto-Sivashinsky equation.**

where

$$
\begin{cases}
a(x,y) = -\frac{\nu_t}{\nu_{t,max}}\frac{y}{r}, \\
b(x,y) = \frac{\nu_t}{\nu_{t,max}}\frac{x}{r}, \\
\nu_t = \mathrm{sech}^2(r)\tanh(r), \\
r = \sqrt{x^2+y^2}, \\
\nu_{t,max} = 0.385,
\end{cases}
$$

and $u_{\mathrm{ic}}$, $u_{\mathrm{bc}}$ specify the initial condition and the boundary condition, respectively. The exact solution is given by

$$
u(x,y,t) = -\tanh(\frac{y}{2}\cos(\omega t) - \frac{x}{2}\sin(\omega t)),
$$

where $\omega = \frac{1}{r}\frac{\nu_t}{\nu_{t,max}}$, and one can derive the initial condition and the boundary condition from the exact solution.

Although the governing equation is linear, the problem remains challenging because the spatially varying rotational field induces strong shear, leading to rapid spectral enrichment, increasingly oscillatory solution structure, and pronounced multi-scale behavior. These effects manifest as thin, dynamically evolving filaments with large gradients, which are difficult for standard PINNs to resolve. In addition, the PDE is non-separable, so the space-time variables are strongly coupled rather than well approximated by a simple separated representation.

We apply TINN of 1185 parameters with its backbone structure, $2 \times 20 \times 20 \times 1$, and its layer-wise time-embedding network, $1 \times 10 \times 10 \times 5$. Under 1250 iterations of LM optimizer, we get the relative $L^2$ error $9.45E - 04$. The ground truth, the network prediction, and the error at different time-slices are displayed in Figure 8.

**B.8. $(3+1)$-D Klein-Gordon Equation**

We consider the three-dimensional Klein-Gordon equation on the space-time domain $[-1,1] \times [-1,1] \times [-1,1] \times [0,10]$. The solution $u(x,y,z,t)$ satisfies

$$
\begin{cases}
u_{tt} - \Delta u + u^2 = f, & (x,y,z,t) \in (-1,1) \times (-1,1) \times (-1,1) \times (0,10), \\
u(x,y,z,0) = x+y+z, & (x,y,z) \in [-1,1] \times [-1,1] \times [-1,1], \\
u_t(x,y,z,0) = 2xyz, & (x,y,z) \in [-1,1] \times [-1,1] \times [-1,1], \\
u(x,y,z,t) = u_{\mathrm{bc}}(x,y,z,t), & (x,y,z,t) \in \partial([-1,1] \times [-1,1] \times [-1,1]) \times [0,10],
\end{cases}
$$

where $f$ denotes a source term and $u_{\mathrm{bc}}$ specifies the boundary condition. The exact solution is given by

$$
u(x,y,z,t) = (x+y+z)\cos(2t) + xyz\sin(2t),
$$

from which both the source term $f$ and the boundary data $u_{\mathrm{bc}}$ are derived.

We apply TINN of 1225 parameters with its backbone structure, $3 \times 20 \times 20 \times 1$, and its layer-wise time-embedding network, $1 \times 10 \times 10 \times 5$. Under 10000 iterations of LM optimizer, we get the relative $L^2$ error $5.20E - 05$.

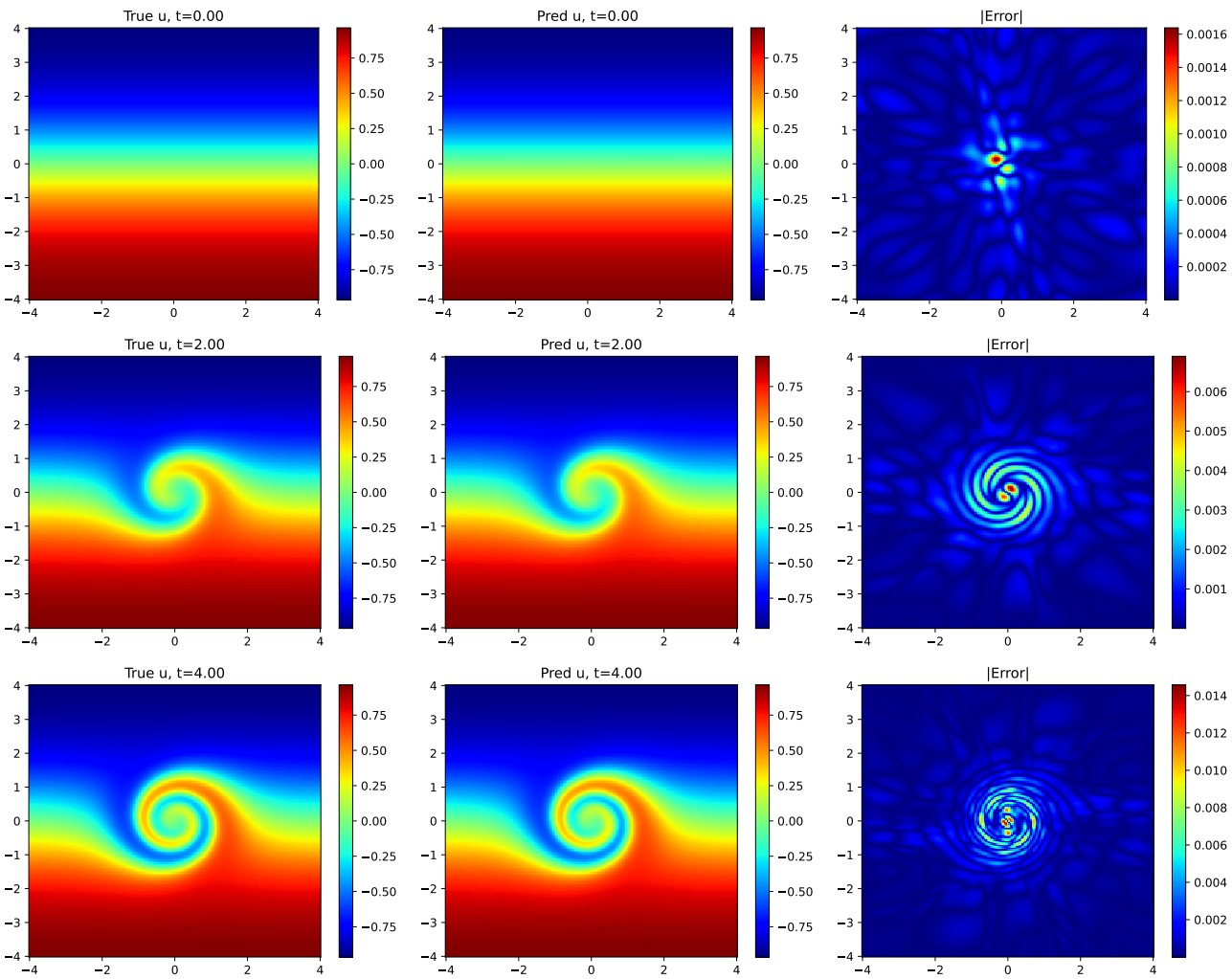

*Figure 8.* **Flow Mixing solution.** We plot the ground truth, the network prediction, and the error at $t = 0, \ 2, \ 4$.

### B.9. $(3+1)$-D Navier-Stokes Equation

We consider the three-dimensional vorticity form of Navier-Stokes equation on the space-time domain $[0, 2\pi] \times [0, 2\pi] \times [0, 2\pi] \times [0, 5]$. The solution $u(x, y, z, t)$ satisfies

$$
\begin{cases}
\boldsymbol{\omega}_t + (\boldsymbol{u} \cdot \nabla)\boldsymbol{\omega} = (\boldsymbol{\omega} \cdot \nabla)\boldsymbol{u} + \nu\Delta\boldsymbol{\omega} + \boldsymbol{f}, & (x, y, z, t) \in (0, 2\pi) \times (0, 2\pi) \times (0, 2\pi) \times (0, 5), \\
\nabla \cdot \boldsymbol{u} = 0, & (x, y, z, t) \in (0, 2\pi) \times (0, 2\pi) \times (0, 2\pi) \times (0, 5), \\
\boldsymbol{\omega}(x, y, z, 0) = \boldsymbol{\omega}_0(x, y, z), & (x, y, z) \in [0, 2\pi] \times [0, 2\pi] \times [0, 2\pi], \\
\boldsymbol{u}(x, y, z, 0) = \boldsymbol{u}_0(x, y, z), & (x, y, z) \in [0, 2\pi] \times [0, 2\pi] \times [0, 2\pi], \\
\boldsymbol{\omega}(x, y, z, t) = \boldsymbol{\omega}_{\text{bc}}(x, y, z, t), & (x, y, z, t) \in \partial([0, 2\pi] \times [0, 2\pi] \times [0, 2\pi]) \times [0, 5],
\end{cases}
$$

where $\boldsymbol{f}$ denotes a vector source term, $\boldsymbol{\omega}_0$ and $\boldsymbol{u}_0$ specify the initial conditions, and $\boldsymbol{\omega}_{\text{bc}}$ specifies the boundary condition. The exact solution is given by

$$
\begin{cases}
u^x = 2e^{-9\nu t}\cos(2x)\sin(2y)\sin(z), \\
u^y = -e^{-9\nu t}\sin(2x)\cos(2y)\sin(z), \\
u^z = -2e^{-9\nu t}\sin(2x)\sin(2y)\cos(z), \\
\omega^x = -3e^{-9\nu t}\sin(2x)\cos(2y)\cos(z), \\
\omega^y = 6e^{-9\nu t}\cos(2x)\sin(2y)\cos(z), \\
\omega^z = -6e^{-9\nu t}\cos(2x)\cos(2y)\sin(z),
\end{cases}
$$

where $\nu$ is 0.05, both the source term $\boldsymbol{f}$ and the boundary data $\boldsymbol{\omega}_{\mathrm{bc}}$ can be derived from the exact solution.

We apply TINN of 1305 parameters with its backbone structure, $3 \times 20 \times 20 \times 3$, where the output is for velocity, and one can derive vorticity by $\boldsymbol{\omega} = \nabla \times \boldsymbol{u}$. Regarding to the layer-wise time-embedding network, the structure is $1 \times 10 \times 10 \times 5$. Under 5000 iterations of LM optimizer, we get the relative $L^2$ error $4.05E - 04$.

### B.10. Inverse Problem of Viscous Burgers Equation

We consider the inverse problem of the one-dimensional viscous Burgers equation on the space-time domain $[-1, 1] \times [0, 1]$. We aim to find out parameters $\lambda_1$ and $\lambda_2$ in

$$u_t + \lambda_1 u u_x - \lambda_2 u_{xx} = 0,$$

where the exact $\lambda_1 = 1$ and $\lambda_2 = \frac{0.01}{\pi}$. With 2000 observations, TINN recovers the coefficients with absolute errors $9.25E - 09$ for $\lambda_1$ and $7.72E - 09$ for $\lambda_2$, where TINN uses 1147 parameters (two of them is $\lambda_1$ and $\lambda_2$) with its backbone structure $1 \times 20 \times 20 \times 1$, and its layer-wise time-embedding network, $1 \times 10 \times 10 \times 5$.

## C. TINNs: A General Remedy for Time Entanglement

### C.1. Time-Entanglement Problem for Other Architectures

We introduced the *time-entanglement problem* using multilayer perceptrons (MLPs) as an illustrative example. However, this issue is not specific to MLPs and can arise in a broad class of neural architectures when time enters only through input conditioning while the internal representation remains shared across all temporal regimes. In this section, we briefly discuss several representative alternatives.

Consider a one-block ResNet (He et al., 2016) in one spatial dimension,

$$u_{\boldsymbol{\theta}}(x, t) := U(wx + vt + b) + \alpha^x x + \alpha^t t.$$

The spatial derivative is

$$\partial_x u_{\boldsymbol{\theta}}(x, t) = U'(wx + vt + b) \, w + \alpha^x,$$

which introduces an additional affine degree of freedom through $\alpha^x$ compared to a vanilla MLP. While this increases expressivity, temporal variation still acts only through a shift in the argument of $U'$, and the underlying spatial representation remains fixed across time. As a result, the time-entanglement problem persists.

Next, consider a modified MLP (Wang et al., 2021) of the form

$$u_{\boldsymbol{\theta}}(x, t) := U(wx + vt + b) \, \sigma(w_1 x + v_1 t + b_1) + \big(1 - U(wx + vt + b)\big) \, \sigma(w_2 x + v_2 t + b_2).$$

The resulting spatial derivative is more structured and can partially alleviate interference between temporal regimes. Nevertheless, temporal dependence still enters exclusively through shifts in activation arguments, while the functional form of the spatial representation remains unchanged. Consequently, temporal evolution cannot be represented as flexibly as in TINNs.

Finally, consider separable physics-informed neural networks (SPINNs) (Cho et al., 2023), in which

$$u_{\boldsymbol{\theta}}(x, t) := U(x) \, V(t).$$

Here,

$$\partial_x u_{\boldsymbol{\theta}}(x, t) = U'(x) \, V(t),$$

allowing time-dependent scaling of spatial features. While this avoids pure activation shifts, the spatial representation $U'(x)$ itself remains time-independent and cannot adapt its structure as the solution evolves.

In contrast, TINNs explicitly parameterize temporal evolution as a trajectory in parameter space, enabling the spatial representation itself to change over time. This removes the shared-representation constraint underlying time entanglement and yields a more general and flexible modeling framework for time-dependent PDEs.

## C.2. TINNs Generalize to Other Backbones

The TINN framework is not restricted to multilayer perceptrons as spatial backbones. More generally, TINNs incorporate temporal dependence by parameterizing the network weights as smooth functions of time, rather than treating time as an additional input coordinate. This construction can be applied to a wide range of neural architectures.

For example, given a ResNet (He et al., 2016), a time-induced formulation takes the form

$$u_{\boldsymbol{\theta}(t)}(x) := U(w(t)x + b(t)) + \alpha^x(t)x,$$

where all parameters evolve with time. Compared to a space–time ResNet with fixed weights, this allows the spatial representation itself to adapt dynamically.

Similarly, for a modified MLP (Wang et al., 2021), we may define

$$u_{\boldsymbol{\theta}(t)}(x) := U(w(t)x + b(t))\sigma(w_1(t)x + b_1(t)) + (1 - U(w(t)x + b(t)))\sigma(w_2(t)x + b_2(t)),$$

again allowing all parameters to vary with time. This yields a time-adaptive mixture structure while preserving the original architectural design.

For SPINNs (Cho et al., 2023), temporal dependence is already factorized from spatial variables. In the one-dimensional spatial case, applying TINNs recovers the same effective representation as using an MLP backbone with time-dependent parameters. In higher spatial dimensions, e.g., $d = 2$, a natural extension is

$$u_{\boldsymbol{\theta}(t)}(x, y) := U_{\boldsymbol{\theta_1}(t)}(x)U_{\boldsymbol{\theta_2}(t)}(y),$$

where each spatial component evolves independently over time.

These examples illustrate that the core idea of TINNs—representing temporal evolution as a trajectory in parameter space—is architecture-agnostic. As a result, TINNs can be readily integrated with a broad class of neural backbones beyond standard MLPs.

## D. LM optimizer

For clarity, we first describe the Levenberg–Marquardt (LM) method for a single least-squares loss term arising from a PDE residual. Consider

$$\mathcal{L}u_{\boldsymbol{\psi}} = f,$$

where $\mathcal{L}$ is a (possibly nonlinear) differential operator, $f$ is a given source term, and $u_{\boldsymbol{\psi}}$ denotes a neural network parameterized by $\boldsymbol{\psi}$.

Let $\{\mathbf{x}_i\}_{i=1}^N$ be a set of collocation points. Define the residual vector $\mathbf{L} \in \mathbb{R}^N$ and the Jacobian matrix $\mathbf{J} \in \mathbb{R}^{N \times P}$ by

$$\mathbf{J} = \begin{bmatrix} \frac{1}{\sqrt{N}}\nabla_{\boldsymbol{\psi}}\mathcal{L}u_{\boldsymbol{\psi}}(\mathbf{x}_1) \\ \vdots \\ \frac{1}{\sqrt{N}}\nabla_{\boldsymbol{\psi}}\mathcal{L}u_{\boldsymbol{\psi}}(\mathbf{x}_N) \end{bmatrix}, \quad \mathbf{L} = \begin{bmatrix} \frac{1}{\sqrt{N}}(\mathcal{L}u_{\boldsymbol{\psi}}(\mathbf{x}_1) - f(\mathbf{x}_1)) \\ \vdots \\ \frac{1}{\sqrt{N}}(\mathcal{L}u_{\boldsymbol{\psi}}(\mathbf{x}_N) - f(\mathbf{x}_N)) \end{bmatrix},$$

where $P$ is the number of trainable parameters.

With this notation, the loss can be written compactly as

$$\ell(\boldsymbol{\psi}) = \frac{1}{2}\|\mathbf{L}\|^2.$$

When multiple loss components are present—such as PDE residuals, initial conditions, and boundary conditions—the corresponding residual vectors and Jacobians are stacked vertically to form a single augmented least-squares system.

At each iteration, the LM method computes an update $\delta\boldsymbol{\psi}$ by solving the damped normal equations

$$(\mathbf{J}^\top\mathbf{J} + \mu I)\delta\boldsymbol{\psi} = \mathbf{J}^\top\mathbf{L},$$

where $\mu > 0$ is the damping parameter. The parameter vector is then updated as

$$\boldsymbol{\psi} \leftarrow \boldsymbol{\psi} - \delta\boldsymbol{\psi},$$

with $\mu$ adjusted adaptively according to the standard LM acceptance criterion.

We summarize the complete LM procedure used in our experiments in Algorithm 1.

---

**Algorithm 1** LM Algorithm

---

**Require:** Training epochs $E$; loss construction $\mathcal{L}$ induced by PDE/BC/IC residuals; initial damping $\mu_0$; update factors
$\quad\gamma_\uparrow > 1, \gamma_\downarrow > 1$; bounds $\mu_{\min}, \mu_{\max}$; safeguard factor $\eta \geq 1$ (for updating $\mu_{\min}$)
**Ensure:** Trained network $u_{\boldsymbol{\psi}}(\mathbf{x})$
$\quad$ Initialize $\boldsymbol{\psi} \leftarrow \boldsymbol{\psi}_0$, $\mu \leftarrow \mu_0$, $\ell_{\text{old}} \leftarrow 10^5$
$\quad$ **for** $k = 1$ to $E$ **do**
$\quad\quad$ Assemble residual vector ($\mathbf{L}$), Jacobian ($\mathbf{J}$) and calculate loss ($\ell$)
$\quad\quad$ Proposed update: $\boldsymbol{\psi} \leftarrow \boldsymbol{\psi} - (\mathbf{J}^\top\mathbf{J} + \mu I)^{-1}\mathbf{J}^\top\mathbf{L}$
$\quad\quad$ **if** $\ell < \ell_{\text{old}}$ **then**
$\quad\quad\quad$ $\mu \leftarrow \max(\mu/\gamma_\downarrow, \mu_{\min})$ $\qquad\qquad\qquad\qquad\qquad\qquad\qquad\qquad$ ▷ accept; decrease damping
$\quad\quad$ **else**
$\quad\quad\quad$ $\mu \leftarrow \min(\mu \cdot \gamma_\uparrow, \mu_{\max})$ $\qquad\qquad\qquad\qquad\qquad\qquad\qquad\qquad$ ▷ reject; increase damping
$\quad\quad$ **end if**
$\quad\quad$ $\ell_{\text{old}} \leftarrow \ell$
$\quad\quad$ **if** $\ell/\mu > 10^5$ **then**
$\quad\quad\quad$ $\mu \leftarrow \ell/10$ $\qquad\qquad\qquad\qquad\qquad\qquad\qquad$ ▷ safeguard for ill-conditioning / poor scaling
$\quad\quad\quad$ $\mu_{\min} \leftarrow \eta\,\mu_{\min}$
$\quad\quad$ **end if**
$\quad$ **end for**

---

# E. More Experiment

### E.1. Comparison to the Space–Time Separation Method

We conduct an experiment on a transport equation to compare the time–space separation method (Datar et al., 2026) and TINN. The solution $u(x, t)$ satisfies

$$\begin{cases} u_t - u_x = 0, & (x, y) \in (0, 2) \times (0, 1), \\ u(x, 0) = \ln x, & x \in [0, 2], \\ u(2, t) = \ln(2 + t), & t \in [0, 1], \end{cases}$$

where the exact solution

$$u(x, t) = \ln(x + t).$$

TINN apply 1145 parameters with its backbone structure, $1 \times 20 \times 20 \times 1$, and its layer-wise time-embedding network, $1 \times 10 \times 10 \times 5$.

### E.2. Comparison to Naive Hypernetwork Design

To compare with hypernetwork baselines, we include a new rebuttal experiment against a HyperPINN-style method (de Avila Belbute-Peres et al., 2021) with matched parameter budgets. While HyperPINN globally generates all backbone parameters, TINN uses layer-wise temporal embeddings, providing a more structured and efficient modulation. The result in Table 6 shows that TINN outperforms HyperPINN on Burgers and Klein–Gordon and remains competitive on Allen–Cahn, while requiring less training time, supporting the effectiveness of layer-wise modulation.

### E.3. Distinct Network Structure with Same Training Strategy

To disentangle the effect of the structure and the training strategy, we compare the error performance between different network structure, including MLP, SPINN, PirateNet, and TINN. The optimizer is Adam with the same learning rate

*Table 6.* Relative $L^2$ error comparison between HyperPINN and TINN. The result is the mean of 5 trials (with random seed $0 - 4$).

|  | Rel $L^2$-error | Time | Parameters |
|---|---|---|---|
| **Burgers** | | | |
| HyperPINN | 1.94E-06 $\pm$ 9.34E-07 | 0.90hr | 1162 |
| TINN | **6.89E-07 $\pm$ 3.97E-07** | 0.75hr | 1145 |
| **Allen-Cahn** | | | |
| HyperPINN | **3.49E-06 $\pm$ 1.71E-06** | 0.83hr | 1202 |
| TINN | 3.85E-06 $\pm$ 1.48E-06 | 0.78hr | 1185 |
| **Klein-Gordon** | | | |
| HyperPINN | 2.02E-01 $\pm$ 3.00E-01 | 0.77hr | 1202 |
| TINN | **4.78E-06 $\pm$ 2.63E-06** | 0.67hr | 1185 |

schedule. The result is displayed in Table 7. TINN gets the best performance in the Burgers equation and the Allen-Cahn equation. As in the Klein-Gordon equation, TINN is comparible with other structures with more efficient training budget.

*Table 7.* Relative $L^2$ error comparison between distinct network structure under the same training strategy. The result is the mean of 5 trials (with random seed $0 - 4$).

|  | Rel $L^2$-error | Time | Parameters |
|---|---|---|---|
| **Burgers** | | | |
| MLP | 2.06E-04 $\pm$ 1.72E-04 | 0.89hr | 545340 |
| SPINN | 9.52E-02 $\pm$ 8.16E-02 | 1.02hr | 356716 |
| PirateNet | 1.34E-04 $\pm$ 1.19E-04 | 1.25hr | 532227 |
| TINN | **9.71E-05 $\pm$ 5.38E-05** | 0.44hr | 143811 |
| **Allen-Cahn** | | | |
| MLP | 9.24E-03 $\pm$ 3.20E-03 | 0.68hr | 545340 |
| SPINN | 7.12E-01 $\pm$ 1.94E-01 | 1.22hr | 535074 |
| PirateNet | 7.67E-04 $\pm$ 6.87E-04 | 1.00hr | 532227 |
| TINN | **7.31E-04 $\pm$ 3.51E-04** | 0.34hr | 143811 |
| **Klein-Gordon** | | | |
| MLP | **1.04E-03 $\pm$ 1.99E-04** | 2.13hr | 545340 |
| SPINN | 5.95E-03 $\pm$ 2.89E-03 | 3.26hr | 535074 |
| PirateNet | 1.30E-03 $\pm$ 2.47E-04 | 2.37hr | 532227 |
| TINN | 1.61E-03 $\pm$ 2.76E-04 | 1.84hr | 143811 |

### E.4. Issue of the Training Precision

The issue of numerical precision for fair comparison is important. In our setting, TINN uses double precision mainly because the LM optimizer requires solving linear systems, for which higher precision is often more stable. To examine whether this choice materially affects the comparison, we conduct additional experiments on the Burgers equation in which strong baselines (CoPINN* and PirateNet) are trained in both single and double precision under otherwise identical settings (Table 8). We find that double precision can provide modest gains in some cases (e.g., CoPINN*), but does not consistently improve performance (e.g., PirateNet), and importantly does not close the gap to TINN. Moreover, double precision introduces substantial computational overhead for large models (up to $\sim 10\times$ slower), making it impractical for standard PINN architectures. Taken together, these results suggest that the observed gains are not primarily due to precision differences, but rather to TINN's compact parameterization, which enables stable and effective use of second-order optimization in a tractable regime.

*Table 8.* Comparison of single precision (SP) and double precision (DP) on the Burgers equation with CoPINN* and PirateNet.

| Case (Burgers) | Rel-$L^2$ Error | Num. of Params. | Time |
|---|---|---|---|
| CoPINN* (Adam; SP) | 1.03E-04 | 336864 | 0.78hr |
| CoPINN* (Adam; DP) | 2.75E-05 | 336864 | 5.92hr |
| PirateNet (SOAP; SP) | 1.46E-06 | 534853 | 1.70hr |
| PirateNet (SOAP; DP) | 3.14E-06 | 534853 | 16.8hr |

### E.5. Convergence Speed

Convergence measured by the number of iterations to reach a target error provides a more controlled comparison. Following this idea, we include a convergence-based evaluation, measuring how many iterations (and corresponding wall-clock time) TINN requires to reach the final accuracy achieved by different baselines. As shown in Table 9 for the Burgers equation, TINN reaches the accuracy of a standard PINN ($4.64 \times 10^{-4}$) within 211 iterations (28.2s), and surpasses PirateNet's reported accuracy ($1.46 \times 10^{-6}$) within 6,462 iterations (709.6s), well before exhausting its full training budget. These results demonstrate that TINN achieves comparable or better accuracy with substantially fewer optimization steps.

*Table 9.* The iteration and the time used by TINN to reach the results of other baselines.

| Case: Burgers | Rel-$L^2$ Error | TINN's Iteration | TINN's Time |
|---|---|---|---|
| PINN | 4.64E-04 | 211 | 28.2s |
| CoPINN | 1.03E-04 | 454 | 54.7s |
| PirateNet | 1.46E-06 | 6462 | 709.6s |

### E.6. TINN with Different Optimizers

We include additional results for a small-scale TINN (1145 parameters) trained with different optimizers. InBurgers equation, TINN remains fully trainable with Adam, achieving a relative $L^2$ error of $2.74 \times 10^{-3}$, confirming that its effectiveness does not rely on increased model size. At the same time, higher-order methods such as L-BFGS and LM further improve accuracy under the same architecture, indicating that TINN benefits from stronger optimization while remaining compatible with first-order methods.

*Table 10.* The error performance of TINN with small structure under Adam, L-BFGS, and LM. The results are the mean of 5 trials (with random seed 0–4).

| Case | Rel $L^2$-error | Parameters |
|---|---|---|
| **Burgers** | | |
| TINN + Adam | 3.79E-03 $\pm$ 8.27E-04 | 1145 |
| TINN + L-BFGS | 5.87E-04 $\pm$ 4.53E-04 | 1145 |
| TINN + LM | **6.89E-07 $\pm$ 3.97E-07** | 1145 |
| **Allen-Cahn** | | |
| TINN+ Adam | 1.19E-01 $\pm$ 4.84E-02 | 1185 |
| TINN + L-BFGS | 1.09E-02 $\pm$ 1.12E-02 | 1185 |
| TINN + LM | **3.85E-06 $\pm$ 1.48E-06** | 1185 |
| **Klein-Gordon Equation** | | |
| TINN + Adam | 2.12E-02 $\pm$ 1.23E-02 | 1185 |
| TINN + L-BFGS | 1.26E-02 $\pm$ 9.93E-03 | 1185 |
| TINN + LM | **4.78E-06 $\pm$ 2.63E-06** | 1185 |

## F. Implementation Details and Experimental Settings

### F.1. Comparison between MLP and TINN (Figure 3)

When introducing the Time-Induced Neural Network (TINN) in Section 3, we compared the ability of a vanilla MLP and a TINN to represent spatial derivatives of the solution to the Burgers equation. As shown in Figure 3, the two models are constructed with comparable numbers of parameters to ensure a fair comparison (MLP: 1160; TINN: 1145).

The MLP consists of two hidden layers with 20 and 50 neurons, respectively. In contrast, TINN employs a spatial network with two hidden layers of 20 neurons each, while temporal dependence is introduced through time-varying parameters.

To make the parameter count explicit, we detail the construction used for the Burgers equation in Table 1. The spatial backbone network has architecture $1 \times 20 \times 20 \times 1$ and does not include an output bias. This corresponds to $L = 3$ layers and a time-dependent feature vector $\mathbf{\Phi}(t) \in \mathbb{R}^{2L-1} = \mathbb{R}^5$, with a total of $N_D = 480$ backbone parameters.

The temporal network $\mathcal{N}(t)$ is instantiated as $1\times10\times10\times5$ (again without an output bias), contributing 180 parameters. In addition, we introduce a gating vector $\boldsymbol{\alpha} \in \mathbb{R}^5$.

The total number of trainable parameters in TINN is therefore

$$2N_D + 180 + 5 = 1145,$$

where $2N_D$ arises from the affine lift $\mathbf{F}$, 180 from $\mathcal{N}(t)$, and 5 from $\boldsymbol{\alpha}$.

For comparison, consider a naive parameterization of $\boldsymbol{\theta}(t)$ using the same spatial backbone with $N_D = 480$. If $\boldsymbol{\theta}(t)$ is generated by a fully connected network of architecture $1\times10\times10\times480$, sharing the same hidden width ($h = 10$) as $\mathcal{N}(t)$, the total number of trainable parameters increases to 4930. This comparison illustrates how TINN introduces temporal flexibility with substantially fewer parameters than a naive time-dependent parameterization.

### F.2. Evaluation Metric

To quantitatively compare error performance across methods, we report the relative $L^2$-error, defined as

$$\sqrt{\frac{\sum_{i=1}^{N_{\text{test}}}(u_{\boldsymbol{\theta}(t_i)}(\mathbf{x}_i) - u^*(\mathbf{x}_i, t_i))^2}{\sum_{i=1}^{N_{\text{test}}} u^*(\mathbf{x}_i, t_i)^2}},$$

where $u_{\boldsymbol{\theta}(t)}(\mathbf{x})$ denotes the neural network prediction and $u^*(\mathbf{x}, t)$ denotes the reference solution. The test set $\{(\mathbf{x}_i, t_i)\}_{i=1}^{N_{\text{test}}} \subset \Omega \times [0, T]$ is sampled independently of the training data.

To quantify relative error performance gains, we report the error performance improvement (IMP):

$$\text{acc. IMP} = \frac{e_{\text{baseline}}}{e_{\text{TINN}}},$$

where $e_{\text{TINN}}$ denotes the relative $L^2$-error achieved by the proposed method, and $e_{\text{baseline}}$ denotes the relative $L^2$-error among all competing methods excluding TINNs.

### F.3. Implementation Details for Main Results (Table 2)

We adopt the experimental settings proposed in prior works whenever they address the same benchmark PDEs, and apply minor refinements when necessary to ensure fair and stable comparisons. For example, for CoPINN, which reports results on the Klein–Gordon equation, we follow their original setup and apply additional implementation refinements, achieving a relative error of $10^{-6}$ compared to the $10^{-4}$ reported in their paper. For PirateNet, we use the hyperparameter configurations provided by the authors for the Burgers, Allen–Cahn, Kortweg–De Vries, and wave equations. Due to hardware constraints, the number of neurons is reduced in our implementation; nevertheless, the resulting models still contain substantially more parameters than the other baselines. All experiments are conducted on an NVIDIA A6000 GPU.

Table 11 summarizes the network architectures, training iterations, optimizers, and optimizer-specific hyperparameters used in all experiments. When using Adam, we apply a learning-rate decay schedule specified by the tuple (learning rate, decay rate, warmup, decay step); the same parameterization is adopted for SOAP. For the Levenberg–Marquardt (LM) optimizer, the damping parameter $\mu$ is adjusted dynamically during training according to the configuration ($\mu_0, \gamma_\uparrow, \gamma_\downarrow, \mu_{\max}, \mu_{\min}, \eta$). The update mechanism is described in detail in Appendix D.

All methods except TINN involve a large number of parameters; consequently, these models are trained in single precision, consistent with the original implementations. In contrast, TINN employs a more compact parameterization, and LM training requires solving linear systems; therefore, we train TINN in double precision for improved numerical stability.

All experiments are repeated using five random seeds $\{0, 1, 2, 3, 4\}$. For PINNs- and TINN-based methods, we monitor a validation loss to detect overfitting. Specifically, if the validation loss exceeds five times the training loss, the collocation points in the training dataset are resampled.

In Table 12, we report the number of training points used for each method and PDE. The number of collocation points for the PDE residual is denoted by $N_c$, while $N_{\text{ic}}$ and $N_{bc}$ correspond to the initial and boundary conditions, respectively. Entries marked as N/A indicate that the corresponding training points are not required. For example, the Allen–Cahn and

*Table 11.* Training configurations for each method and PDE.

| Case | Main Structure | iteration | Optimizer | Hyperparameters |
|---|---|---|---|---|
| **Burgers** | | | | |
| PINN | Layer: 4, Neuron: 320 | 250K | Adam | $(10^{-3}, 0.9, 10000, 5000)$ |
| CoPINN* | Layer: 5, Neuron: 200 | 1050K | Adam | $(10^{-3}, 0.9, 10000, 21000)$ |
| PirateNet SOAP | Block: 3, Neuron: 200 | 100K | SOAP | $(10^{-3}, 0.9, 5000, 2000)$ |
| TINN | Layer: 2, x-neu: 20, t-neu: 10 | 30K | LM | $(10, 1.7, 1.3, 10^8, 10^{-12}, 1.0)$ |
| **Allen-Cahn** | | | | |
| PINN | Layer: 4, Neuron: 320 | 250K | Adam | $(10^{-3}, 0.9, 10000, 5000)$ |
| CoPINN* | Layer: 5, Neuron: 200 | 1050K | Adam | $(10^{-3}, 0.9, 10000, 21000)$ |
| PirateNet SOAP | Block: 3, Neuron: 200 | 100K | SOAP | $(10^{-3}, 0.9, 5000, 2000)$ |
| TINN | Layer: 2, x-neu: 20, t-neu: 10 | 30K | LM | $(10, 1.7, 1.3, 10^8, 10^{-12}, 1.0)$ |
| **Korteweg-De Vries** | | | | |
| PINN | Layer: 4, Neuron: 320 | 250K | Adam | $(10^{-3}, 0.9, 10000, 5000)$ |
| CoPINN* | Layer: 5, Neuron: 200 | 1050K | Adam | $(10^{-3}, 0.9, 10000, 21000)$ |
| PirateNet SOAP | Block: 3, Neuron: 200 | 100K | SOAP | $(10^{-3}, 0.9, 5000, 2000)$ |
| TINN | Layer: 2, x-neu: 20, t-neu: 10 | 25K | LM | $(10, 1.7, 1.3, 10^8, 10^{-12}, 1.0)$ |
| **Klein-Gordon** | | | | |
| PINN | Layer: 4, Neuron: 320 | 125K | Adam | $(10^{-3}, 0.9, 10000, 2500)$ |
| CoPINN* | Layer: 5, Neuron: 128 | 200K | Adam | $(10^{-3}, 0.9, 10000, 4000)$ |
| PirateNet SOAP | Block: 3, Neuron: 150 | 100K | SOAP | $(10^{-3}, 0.9, 5000, 2000)$ |
| TINN | Layer: 2, x-neu: 20, t-neu: 10 | 10K | LM | $(10, 1.7, 1.3, 10^8, 10^{-12}, 1.0)$ |
| **Wave** | | | | |
| PINN | Layer: 4, Neuron: 320 | 250K | Adam | $(10^{-3}, 0.9, 10000, 5000)$ |
| CoPINN* | Layer: 5, Neuron: 200 | 1100K | Adam | $(10^{-3}, 0.9, 10000, 22000)$ |
| PirateNet SOAP | Block: 3, Neuron: 200 | 100K | SOAP | $(10^{-3}, 0.9, 5000, 2000)$ |
| TINN | Layer: 2, x-neu: 20, t-neu: 10 | 30K | LM | $(10, 1.27, 1.3, 10^8, 5 \times 10^{-7}, 2.0)$ |

Korteweg–De Vries equations are equipped with periodic boundary conditions; therefore, we employ embedding techniques to enforce these constraints exactly, eliminating the need for boundary collocation points when minimizing the boundary loss. For CoPINN* and PirateNet, we use the same training-point configurations as reported in CoPINN (Duan et al., 2025) and PirateNet (Wang et al., 2025), respectively.

In all experiments that use the Levenberg–Marquardt optimizer, we apply penalty weights to balance the different loss components. Let $\lambda_r$, $\lambda_{ic}$, and $\lambda_b$ denote the penalty terms for the PDE residual, initial condition, and boundary condition, respectively. Specifically, we set $\lambda_{ic}^{-1}$ of the initial-condition function values (average of $60\%$ small values) to capture the small dynamics effectively. The resulting penalty settings for each PDE are listed in Table 13.

For PINN, we use uniform penalty weights, i.e., $\lambda_* = 1$. In CoPINN, although a pointwise "difficulty" measure is computed for each training point, the corresponding penalty weights are also fixed to one. In contrast, PirateNet balances the loss components by normalizing their norms, which leads to penalty weights that vary dynamically during training.

In Section 4, we claim that TINN reaches comparable or better accuracy with a $10.55\times$ speedup over PirateNet with SOAP on the Burgers equation. Specifically, TINN uses $0.16$hr to achieve the accuracy $1.46E - 06$, which is the error performance of the PirateNet with SOAP. On the other hand, TINN uses $0.65$hr to achieve the accuracy $4.72E - 06$, which is the error performance of the PirateNet with SOAP (see in Table 14).

### F.4. Setting for Ablation Studies (Tables 3 and 4)

**Adam Ablation: Impact of Model Capacity.** Table 15 reports the experimental configurations and the standard deviation of the relative $L^2$-error across trials for TINNs with large numbers of parameters trained using the Adam optimizer. In the network specification, $\{n\}^4$ denotes four hidden layers with $n$ neurons each, while the leading number indicates the dimension of the random Fourier feature embedding (e.g., 256 or 50).

Following PirateNet, we employ parameter initialization and causal training for TINN. During training, the penalty terms are

*Table 12.* Number of training points for each method and PDE.

| Case | $N_c$ | $N_{\text{ic}}$ | $N_{bc}$ |
|---|---|---|---|
| **Burgers** | | | |
| PINN | 10000 | 500 | 400 |
| CoPINN* | 65536 | 256 | 512 |
| PirateNet SOAP | 8192 | 256 | 200 |
| TINN | 10000 | 500 | 400 |
| **Allen-Cahn** | | | |
| PINN | 10000 | 500 | N/A |
| CoPINN* | 65536 | 256 | N/A |
| PirateNet SOAP | 8192 | 512 | N/A |
| TINN | 10000 | 500 | N/A |
| **Korteweg-De Vries** | | | |
| PINN | 10000 | 500 | N/A |
| CoPINN* | 65536 | 256 | N/A |
| PirateNet SOAP | 8192 | 512 | N/A |
| TINN | 10000 | 500 | N/A |
| **Klein-Gordon** | | | |
| PINN | 15000 | 4000 | 8000 |
| CoPINN* | 16777216 | 65536 | 262144 |
| PirateNet SOAP | 8192 | 10201 | 81204 |
| TINN | 15000 | 4000 | 8000 |
| **Wave** | | | |
| PINN | 10000 | 500 | 400 |
| CoPINN* | 65536 | 256 | 512 |
| PirateNet SOAP | 8192 | 128 | 400 |
| TINN | 10000 | 500 | 400 |

*Table 13.* Penalty weights for each loss component in TINN training

| Case | $\lambda_{\mathbf{r}}$ | $\lambda_{\text{ic}}$ | $\lambda_b$ |
|---|---|---|---|
| Burgers | 1 | 2 | 1 |
| Allen-Cahn | 1 | 20 | N/A |
| Korteweg-De Vries | 1 | 2 | N/A |
| Klein-Gordon | 1 | 3 | 1 |
| Wave | 1 | 2 | 10 |

*Table 14.* Comparison for the convergence speed between PirateNet and TINN. The error threshold is the error performance obtained from PirateNet with SOAP. All of the experiments are with random seed 4.

| Case | Error Threshold | PirateNet | TINN | Speed Improvement |
|---|---|---|---|---|
| Burgers | 1.46E-06 | 1.70hr | 0.16hr | $10.55\times$ |
| Allen-Cahn | 4.72E-06 | 1.50hr | 0.65hr | $2.30\times$ |

dynamically adjusted by normalizing the gradient magnitudes, as proposed in (Wang et al., 2024a). Since the total number of training iterations differs between the two methods, the penalty normalization is updated every 1000 iterations for PirateNet and every 1500 iterations for TINN. All experiments in this subsection are conducted using single-precision arithmetic.

**LM Ablation: Impact of Network Structure.** Table 16 compares the performance of three network architectures—two MLP-based models and one TINN—trained using the LM optimizer. subMLP is the special case of TINN sharing the same backbone. MLP shares similar backbone (two hidden layers with 20 neurons in the first hidden layer) with TINN, and use comparible number of parameters. We report the mean and standard deviation of the relative $L^2$-error across multiple trials. All experiments are conducted in double precision.

*Table 15.* Experimental settings and performance of large-capacity TINNs and PirateNet trained using the Adam optimizer.

| Case | Rel $L^2$-Error | Iteration | Hyperparameters | $t$-structure | Main structure |
|------|-----------------|-----------|-----------------|---------------|----------------|
| **Burgers** | | | | | |
| PirateNet | 2.43E-05 $\pm$ 4.01E-06 | 100K | $(10^{-3}, 0.9, 5000, 2000)$ | – | block: 3, neuron: 200 |
| TINN | **2.15E-05 $\pm$ 3.75E-06** | 160K | $(10^{-3}, 0.9, 10000, 2000)$ | $1 \times \{150\}^4 \times 9$ | $1 \times 256 \times \{150\}^4 \times 1$ |
| **Allen-Cahn** | | | | | |
| PirateNet | 2.32E-03 $\pm$ 9.60E-04 | 100K | $(10^{-3}, 0.9, 5000, 2000)$ | – | block: 3, neuron: 200 |
| TINN | **8.29E-05 $\pm$ 2.80E-05** | 160K | $(10^{-3}, 0.9, 10000, 2000)$ | $1 \times \{150\}^4 \times 9$ | $1 \times 2 \times 256 \times \{150\}^4 \times 1$ |
| **Klein-Gordon** | | | | | |
| PirateNet | 5.60E-05 $\pm$ 5.11E-06 | 100K | $(10^{-3}, 0.9, 5000, 2000)$ | – | block: 3, neuron: 150 |
| TINN | **4.98E-05 $\pm$ 8.31E-06** | 200K | $(10^{-3}, 0.9, 10000, 2000)$ | $1 \times 50 \times \{100\}^4 \times 9$ | $2 \times 50 \times \{100\}^4 \times 1$ |

*Table 16.* Error performance and variability of two MLP architectures and one TINN with similar parameter counts under LM training.

| Case | Rel $L^2$-Error | Iteration | Hyperparameters | $t$-structure | Main structure |
|------|-----------------|-----------|-----------------|---------------|----------------|
| **Burgers** | | | | | |
| subMLP | 7.11E-05 $\pm$ 4.10E-05 | 110K | $(10, 1.7, 1.3, 10^8, 10^{-9}, 1.0)$ | – | $2 \times 20 \times 20 \times 1$ |
| MLP | 1.92E-05 $\pm$ 1.03E-05 | 34K | $(10,1.7,1.3,10^8,10^{-12}, 1.0)$ | – | $2 \times 20 \times 50 \times 1$ |
| PirateNet | 9.17E-07 $\pm$ 9.66E-07 | 30K | $(10,1.7,1.3,10^8,10^{-12}, 1.0)$ | – | block: 3, neuron: 11 |
| TINN | **6.89E-07 $\pm$ 3.97E-07** | 30K | $(10,1.7,1.3,10^8, 10^{-12}, 1.0)$ | $1 \times 10 \times 10 \times 5$ | $1 \times 20 \times 20 \times 1$ |
| **Allen-Cahn** | | | | | |
| subMLP | 1.76E-03 $\pm$ 1.10E-03 | 110K | $(10, 1.7, 1.3, 10^8, 10^{-9}, 1.0)$ | – | $2 \times 3 \times 20 \times 20 \times 1$ |
| MLP | 7.14E-06 $\pm$ 8.57E-07 | 34K | $(10,1.7,1.3,10^8, 10^{-12}, 1.0)$ | – | $2 \times 3 \times 20 \times 51 \times 1$ |
| PirateNet | 4.09E-05 $\pm$ 5.88E-05 | 30K | $(10,1.7,1.3,10^8,10^{-12}, 1.0)$ | – | block: 3; neuron: 11 |
| TINN | **3.85E-06 $\pm$ 1.48E-06** | 30K | $(10,1.7,1.3,10^8, 10^{-12}, 1.0)$ | $1 \times 10 \times 10 \times 5$ | $1 \times 2 \times 20 \times 20 \times 1$ |
| **Klein-Gordon** | | | | | |
| subMLP | 2.84E-05 $\pm$ 1.24E-05 | 35K | $(10, 1.7, 1.3, 10^8, 10^{-8}, 1.0)$ | – | $3 \times 20 \times 20 \times 1$ |
| MLP | 5.11E-06 $\pm$ 3.87E-06 | 11.1K | $(10,1.7,1.3,10^8, 10^{-12}, 1.0)$ | – | $3 \times 20 \times 51 \times 1$ |
| PirateNet | 8.16E-01 $\pm$ 4.06E-01 | 10K | $(10,1.7,1.3,10^8,10^{-12}, 1.0)$ | – | block: 3; neuron: 11 |
| TINN | **4.78E-06 $\pm$ 2.63E-06** | 10K | $(10,1.7,1.3,10^8, 10^{-12}, 1.0)$ | $1 \times 10 \times 10 \times 5$ | $2 \times 20 \times 20 \times 1$ |

