# OpenReview forum: "TINNs: Time-Induced Neural Networks for Solving Time-Dependent PDEs"
_ICML.cc/2026/Conference — ICML 2026 regular_

### Official Review · Reviewer_Ydz2 · 2026-02-28

**Soundness:** 2
**Presentation:** 3
**Significance:** 2
**Originality:** 2
**Overall Recommendation:** 4
**Confidence:** 4

**Summary:**

This paper presents a PINN variant that aims to stabilize the training and improve accuracy. The standard PINN methods suffer from a time-entanglement issue since a single shared representation struggles to model evolving spatial scales. The proposed method incorporates the time through an independent network and improves the performance of PINNs on learning complex dynamics. This paper also introduces a fast training scheme via Levenberg–Marquardt. The results have shown the improvement of computational efficiency and solution accuracy of the proposed method compared to baseline models across several low-dimensional benchmark PDEs.

**Compliance With Llm Reviewing Policy:**

Affirmed.

**Final Justification:**

The authors have addressed my concerns about how to differentiate their work from related work, baselines, and other empirical experiments. This paper incorporates temporal inductive bias into the PINN pipeline, which is interesting. But this paper needs to be revised to fit into this specific niche.

**Key Questions For Authors:**

- A follow-up question regarding the scalability is the Levenberg–Marquardt (LM) method used in this paper. This paper shows greater efficiency, which could mainly be due to the use of the LM method in a small-model regime. In other words, to handle higher-dimensional or higher-resolution PDEs, we typically use a larger network. In such cases, is the LM method still applicable here?


- I have a few questions regarding the ablation study. The first ablation study is about the optimization method. I am not sure why you compare TINN + Adam with PirateNet. Shouldn't you compare TINN + Adam with TINN + LM? Also, the authors might consider L-BFGS as a baseline. Second, I am also not sure about what MLP is in Table 4. The paper introduced subMLP, but I didn't catch the MLP. Also, what if you have a more powerful network, such as attention layers?

**Limitations:**

No. The authors are encouraged to add a discussion regarding the limitations of the proposed method.

**Strengths And Weaknesses:**

## Strengths

- This paper is generally easy to follow and understand.
- This paper tackles an important problem in PINNs regarding multi-scale learning, computational efficiency, and trainability.
- The Levenberg–Marquardt method fits well to the small-parameter regime and is explained clearly.
- The empirical results are reported within their chosen scope and show the superiority of the proposed method in terms of efficiency and accuracy.

## Weaknesses

- My first concern is regarding the time-decomposed setup. This is not a new topic, as also shown in separable PINNs [1]. This paper designs a different network to tackle the temporal learning part and also uses a different optimization method. I would suggest adding separable PINNs as one of the baseline methods, and discussing the differences and similarities of separable PINNs versus the proposed method.

- The discussion on related work is not sufficient. There is a bunch of PINN-related papers discussing why PINNs are difficult to train [2,3,4]. The authors need to have a more comprehensive analysis of the prior work and compare it to the proposed method. It is not clear what the advantage of the proposed method is.

- It would be good to test the scalability of the proposed method on higher-dimensional PDEs or higher-resolution grids. The current experiments are kind of limited to low-dimensional PDEs with a relatively short time horizon, where the existing PINN family can do a good job. It would be better to have some examples showing what current PINNs cannot do, but your method can.

- The paper would benefit from a deeper theoretical or convergence analysis of the proposed space–time decomposition, which currently appears closely related to classical separation-of-variables methods.

---

Refs:

[1] Cho, Junwoo, et al. "Separable physics-informed neural networks." Advances in Neural Information Processing Systems 36 (2023): 23761-23788.

[2] Krishnapriyan, Aditi, et al. "Characterizing possible failure modes in physics-informed neural networks." Advances in neural information processing systems 34 (2021): 26548-26560.

[3] Wu, Haixu, et al. "Ropinn: Region optimized physics-informed neural networks." Advances in Neural Information Processing Systems 37 (2024): 110494-110532.

[4] Rathore, Pratik, et al. "Challenges in training pinns: A loss landscape perspective." arXiv preprint arXiv:2402.01868 (2024).

---

> ### Author Rebuttal · Authors · 2026-03-31
>
> # Response to Reviewer Ydz2
> We thank Reviewer Ydz2 for the constructive feedback and the opportunity to clarify our work. We address the specific concerns below.
> ### W1: Regarding the relationship with Separable PINNs (SPINN) and CoPINN.
> We thank the reviewer and agree that SPINN is an important baseline. While SPINN uses explicit space–time factorization, TINN employs implicit layer-wise temporal modulation via a hypernetwork, enabling more flexible representations. To directly compare, we include a new rebuttal experiment on the $(3+1)$-D **Klein–Gordon equation**, where TINN achieves $2.36\times10^{-6}$ vs. $9.3\times10^{-3}$ for SPINN, showing substantial improvement. We will expand such comparisons in the camera-ready version. We note that CoPINN builds on SPINN, but does not replace a direct comparison.
> | Method | Rel $L^2$-Error |
> | :--- | :--- |
> | **TINN** | **2.36e-6** |
> | SPINN | 9.3e-3 |
> ### W2: Regarding the discussion on prior work and the advantages of TINN.
> We thank the reviewer for these references and agree that understanding PINN failure modes is crucial. Prior work identifies challenges such as spectral bias, spatial imbalance, and ill-conditioned optimization, focusing on **why** PINNs are difficult to train. In contrast, TINN addresses **how** temporal information is incorporated: instead of joint space–time encoding at the input, it uses layer-wise temporal modulation to adapt intermediate representations over time. This leads to improved trainability and is orthogonal to existing strategies (e.g., spatial decomposition), enabling straightforward combination. We will revise the manuscript to better position TINN relative to these works and their identified failure modes.
> ### W3: Regarding the scalability and complexity of the proposed method.
> We thank the reviewer for this suggestion and include additional rebuttal experiments in challenging regimes where standard PINNs often struggle. On the $(3+1)$-D **KleinGordon equation** ($t\in[0,10]$), TINN achieves a relative $L^2$ error of $2.36\times10^{-6}$ with only 1,225 parameters, outperforming SPINN ($9.3\times10^{-3}$). On the chaotic **Kuramoto-Sivashinsky equation**, TINN achieves $8.58\times10^{-3}$ versus $3.86\times10^{-2}$ for PirateNet. These results demonstrate strong performance in high-dimensional, long-time, and chaotic settings. We will include additional experiments in the camera-ready version.
> ### W4: Regarding the theoretical and convergence analysis of the space–time decomposition.
> We thank the reviewer for this suggestion. While TINN is related in spirit to classical space–time decomposition (e.g., separation of variables), it introduces an implicit, learnable decomposition via layer-wise temporal modulation, rather than explicit factorization with prescribed bases. **Proposition 5.1** shows that TINN strictly generalizes standard space–time MLPs, preserving expressivity while altering how temporal information is incorporated. A rigorous analysis of convergence and conditioning remains an open problem and is not the focus here; instead, we aim to provide a simple and practical approach to mitigate time-entanglement, supported by consistent empirical improvements. We will revise the manuscript to better clarify these distinctions and the scope of our theoretical claims.
> ### Q1: Regarding the scalability and the applicability of the LM method.
> We agree that LM becomes prohibitive for very large networks. However, TINN achieves strong performance with compact parameterization, even in high-dimensional settings, since the hypernetwork output is independent of spatial dimension. For example, on the $(3+1)$-D Klein–Gordon equation, TINN achieves $2.36\times10^{-6}$ with only 1,225 parameters, outperforming SPINN ($9.3\times10^{-3}$), remaining within a tractable LM regime. For larger-scale problems, TINN is fully compatible with first-order optimizers (e.g., Adam), indicating that its benefits stem from the architecture rather than the optimizer.
> ### Q2: Clarification on the ablation studies (Optimizers and Model Architecture).
> We agree that isolating optimizer effects is important. We include additional rebuttal results comparing TINN trained with **Adam, L-BFGS, and LM** under identical settings, showing that while TINN is fully trainable with Adam, higher-order methods consistently improve accuracy. The comparison with PirateNet under Adam evaluates performance in a larger-scale regime under a shared optimizer. Following the reviewer’s suggestion, we explicitly include L-BFGS as a baseline to provide a more complete optimizer study. In Table 4, “MLP” refers to a standard fully-connected network with matched parameters. TINN is a backbone-agnostic **plug-and-play structural prior**, and extending it to more expressive architectures is left for future work.
> | Case | Rel $L^2$ | Parameters |
> | :--- | :--- | :--- |
> | TINN + Adam | 2.74E-03 | 1145 |
> | TINN + L-BFGS | 2.87E-04 | 1145 |
> | TINN + LM | **5.67E-07** | 1145 |

---

> > ### Author Rebuttal · Reviewer_Ydz2 · 2026-04-02
> >
> > Thanks for the rebuttal. My concerns have been addressed. Please revise your paper as promised. I will raise my score to 4.

---

> > > ### Author Response · Authors · 2026-04-03
> > >
> > > We sincerely thank the reviewer for the positive feedback and for raising the score. We are greatly encouraged to hear that our responses have addressed all of your concerns. As promised, we will revise the manuscript accordingly.

---

### Official Review · Reviewer_KAeR · 2026-03-11

**Soundness:** 2
**Presentation:** 3
**Significance:** 3
**Originality:** 4
**Overall Recommendation:** 4
**Confidence:** 4

**Summary:**

TINNs targets the issue that when using PINNs to solve time-dependent PDEs, a single set of parameters has to adapt to different time regimes, which leads to decreased accuracy and unstable training. TINNs uses the time variable together with trainable parameters to generate the layer parameters that process spatial inputs, so that the layer parameters for spatial inputs can evolve over time. Experiments on multiple time-dependent PDEs show that TINNs achieves significant improvements over the baselines.

**Compliance With Llm Reviewing Policy:**

Affirmed.

**Final Justification:**

The method shows a certain degree of innovation, and if the experimental procedure is revised as described in the response, there should be no fundamental issues.

**Key Questions For Authors:**

Why do the models not use exactly the same training strategy?

Why not compare convergence speed by the number of iterations required for the loss to drop below a certain threshold, or for L2-ERROR to drop below a certain threshold? Even if the computation per iteration differs, total FLOPs required to converge can be used to measure convergence speed.

It is not a professional consensus that a model must reach a certain parameter scale to be trainable with Adam; some optimization problems with fewer than 1k parameters are also trained with Adam (Souza et al., 2019; Kumar, 2020). What is the basis for increasing the parameter count of TINNs when training with Adam? Why not present results of training TINNs with Adam under the original parameter count?



Souza, L. O., Ramos, G. O., and Ralha, C. G. Experience Sharing Between Cooperative Reinforcement Learning Agents. arXiv preprint arXiv:1911.02191, 2019.

Kumar, S. Balancing a CartPole System with Reinforcement Learning – A Tutorial. arXiv preprint arXiv:2006.04938, 2020.

**Limitations:**

yes

**Strengths And Weaknesses:**

Strengths

Novel idea: It generates the layer parameters that process spatial inputs from the time variable, addressing the issue that one fixed parameter set must fit different time regimes.

Strong empirical performance: On Burgers, Allen–Cahn, and Klein–Gordon equations, TINNs shows clear advantages in L2-ERROR and convergence speed.

Weaknesses

Imprecise wording: Accuracy is generally discussed for classification tasks. Using “Accuracy” in the abstract may cause ambiguity.

Experimental setup is not rigorous: Using training time to measure the number of iterations to convergence is somewhat imprecise. Even with the same number of iterations, training time can be affected by factors beyond model structure and training strategy. In addition, the training strategies across different models do not seem to be fully consistent. The paper explicitly states: “For PirateNet, we follow the original experimental protocol and use the iteration count reported in the corresponding work, as its training procedure is not directly comparable under a wall-clock–matched setting.”

---

> ### Author Rebuttal · Authors · 2026-03-31
>
> # Response to Reviewer KAeR
>
> We thank Reviewer KAeR for the constructive feedback and the opportunity to clarify our work. We address the specific concerns below.
>
> ### W1: Regarding the imprecise wording.
>
> We thank the reviewer for this observation. We agree that the term “accuracy” is more commonly associated with classification tasks and may be ambiguous in the context of PDE regression. In the final version, we will replace it with more precise terminology, such as “relative $L^2$ error” or “solution error.”
>
> ### W2 + Q1: Regarding the consistency of the training strategy and evaluation metrics.
>
> We thank the reviewer for this important observation regarding experimental consistency. We agree that fully matched training strategies provide the most controlled comparison. In the current version, however, we adopt the original (or best-performing) training protocols for each baseline to ensure that they reach their reported or near-optimal performance, which is a common practice to avoid underestimating strong methods.
>
> In particular, for PirateNet we follow the original training procedure even when it exceeds our nominal training budget, and for CoPINN we increase the number of iterations to ensure convergence under our setup. We acknowledge that this introduces variability across training settings. We will revise the manuscript to clearly distinguish between (i) best-effort comparisons under original protocols and (ii) controlled comparisons with aligned training settings, ensuring a more rigorous and transparent evaluation.
>
> ### Q2: Regarding convergence speed and evaluation metrics
>
> We thank the reviewer for this suggestion and agree that convergence measured by the number of iterations to reach a target error provides a more controlled comparison. Following this idea, we include a convergence-based evaluation in the rebuttal, measuring how many iterations (and corresponding wall-clock time) TINN requires to reach the final accuracy achieved by different baselines.
>
> As shown in the table below for the Burgers equation, TINN reaches the accuracy of a standard PINN ($4.64\times10^{-4}$) within 211 iterations (28.2s), and surpasses PirateNet’s reported accuracy ($1.46\times10^{-6}$) within 6,462 iterations (709.6s), well before exhausting its full training budget. These results demonstrate that TINN achieves comparable or better accuracy with substantially fewer optimization steps. We will include convergence-based metrics (iterations and time, and where feasible FLOPs) in the revised manuscript for a more rigorous comparison.
>
>
>
> | Case: Burgers | Rel $L^2$ | TINN's Iteration | TINN's Time |
> | :--- | :--- | :--- | :--- |
> | PINN | 4.64E-04 | 211 | 28.2s |
> | CoPINN | 1.03E-04 | 454 | 54.7s |
> | PirateNet | 1.46E-06 | 6462 | 709.6s |
>
> ### Q3: Regarding the parameter scale and optimizer choice (Adam vs. LM).
>
> We thank the reviewer for this clarification and agree that Adam is effective for small-scale models; our intention was not to suggest otherwise, and we will revise the wording to avoid this implication. The motivation for increasing the parameter count of TINN under Adam was to demonstrate scalability to larger models relevant for more complex PDEs, rather than to ensure trainability.
>
> To directly address the reviewer’s question, we include additional rebuttal results for a small-scale TINN ($1145$ parameters) trained with different optimizers. We observe that TINN remains fully trainable with Adam, achieving a relative $L^2$ error of $2.74\times10^{-3}$, confirming that its effectiveness does not rely on increased model size. At the same time, higher-order methods such as L-BFGS and LM further improve accuracy under the same architecture, indicating that TINN benefits from stronger optimization while remaining compatible with first-order methods. We will revise the manuscript to clarify this distinction and present these results more explicitly.
>
>
>
> | Case: Burgers | Rel $L^2$-error | Parameters |
> | :--- | :--- | :--- |
> | TINN + Adam | 2.74E-03 | 1145 |
> | TINN + L-BFGS | 2.87E-04 | 1145 |
> | TINN + LM | 5.67E-07 | 1145 |

---

> > ### Author Rebuttal · Reviewer_KAeR · 2026-04-06
> >
> > The main weaknesses and questions have been addressed.

---

> > > ### Author Response · Authors · 2026-04-06
> > >
> > > Thank you very much for the encouraging evaluation and for updating the score. We are glad to know that our responses have adequately addressed the issues you raised. We will revise the paper accordingly in the final version.

---

### Official Review · Reviewer_EJzp · 2026-03-11

**Soundness:** 3
**Presentation:** 3
**Significance:** 3
**Originality:** 4
**Overall Recommendation:** 4
**Confidence:** 2

**Summary:**

This paper argues that standard space-time PINNs suffer from a “time-entanglement” problem, illustrated through a simple toy MLP analysis. To address this, the authors propose Time-Induced Neural Networks (TINNs), which represent the solution as u_{\theta(t)}(x), i.e., a time-indexed family of spatial networks whose parameters evolve with time rather than treating t as an input coordinate. The paper further trains this model with a Levenberg-Marquardt optimizer, exploiting the nonlinear least-squares structure of the PINN objective. On several benchmark PDEs, the paper reports improved empirical performance over the considered baselines.

**Compliance With Llm Reviewing Policy:**

Affirmed.

**Final Justification:**

I believe the authors have provided explanations that address many of my concerns and misunderstandings. Therefore, I consider the increase in score to be justified.

**Key Questions For Authors:**

Q1. Could the authors clarify the rationale behind the choice of the benchmark PDEs and baselines? In particular, what properties of these equations make them especially suitable for evaluating the claimed time-entanglement effect, and how were the chosen baselines selected relative to other nearby approaches?

Q2. The discussion around $\partial_x u(x,t)$ and the time-entanglement problem, including for viscous Burgers’ equation, is interesting, but it currently remains largely illustrative, not rigurous. Could the authors provide a more analytic and mathematcal analysis for why a standard space-time parameterization may struggle when spatial gradients sharpen over time? In addition, if the claimed limitation is structural, it would be helpful to include analogous analysis for $\partial_t u(x,t)$ or mixed derivatives as well.

Q3. Since the paper states that LM can in principle be applied to general PINN parameterizations, could the authors provide additional comparisons using smaller non-TINN models trained with LM? This would help clarify how much of the improvement comes from the proposed parameterization itself, versus from operating in a compact-model regime where LM is practical.

Q4. The proposed compact layer-wise time embedding is interesting, but the explanation in terms of macro-level coherence and micro-level diversity remains somewhat abstract. Could the authors provide a more concrete explanation, ideally with a small worked example showing how the time embedding affects the parameters of different layers over time?

Q5. In Appendix A.2, the paper states that the commonly used initial condition $u(x,0)=x^2\cos(\pi x)$ is incompatible with periodic boundary conditions and introduces an artificial error floor near t=0. Could the authors explain more explicitly which compatibility condition is violated here, and why this leads to the observed error behavior near the initial time?

**Limitations:**

One limitation of the paper is that the empirical comparison is not perfectly clean, since TINN is trained in double precision while the non-TINN baselines are trained in single precision, which can affect numerical stability and final error.

Another limitation is that the theoretical and mechanistic support remains mostly illustrative rather than rigorous. Proposition 5.1 is essentially a simple reparameterization result, and the discussion around the \partial_x u(x,t) analysis—used to motivate the time-entanglement problem- remains intuitive rather than analytically developed.

**Strengths And Weaknesses:**

Strengths
-   Clear main idea:
The proposed idea is clear and easy to follow. The paper first motivates the time-entanglement issue with a simple illustrative example, and then develops the method from a naive hypernetwork-style parameterization of $\theta(t)$ to a compact layer-wise time embedding that substantially reduces parameter cost while preserving temporal flexibility.

-   Strong empirical results on the chosen benchmarks:
 On the reported PDE benchmarks, TINN consistently achieves the best relative $L_2$ error while using far fewer trainable parameters than the baselines. The paper also reports faster convergence than strong baselines such as PirateNet+SOAP on several tasks.

-   Useful ablation studies: The Adam ablation is useful as a first attempt to separate the contribution of the architecture from that of LM, and the LM ablation against subMLP/MLP with similar parameter counts makes the empirical case for the proposed parameterization more convincing.

-   Reports on implementation detail:
The appendix provides substantial detail on architectures, iterations, optimizer settings, training points, and reproducibility-related choices, which improves the paper’s clarity and reproducibility.


Weaknesses
-   The theoretical contribution appears limited:
Proposition 5.1 is technically correct, but it is largely a straightforward reparameterization argument: a standard space–time MLP is recovered by absorbing temporal dependence into the first-layer bias. As a result, this result provides useful intuition, but not a particularly deep theoretical explanation for the observed optimization gains.

-   The theoretical and mechanistic support is limited and mostly heuristic:
Proposition 5.1 is technically correct, but it amounts largely to a straightforward reparameterization argument: a standard space–time MLP can be viewed as a special case of TINN by absorbing temporal dependence into the first-layer bias. Beyond this, the subsequent discussion of optimization flexibility remains suggestive but largely intuitive rather than theoretically sharp. In particular, the toy derivative-based arguments do not provide a strong analysis of approximation, conditioning, or generalization. Appendix B extends the discussion to other architectures, but this part also remains mostly conceptual and does not clearly establish when or why TINN should outperform nearby alternatives.

-   The paper’s positioning against related alternatives remains incomplete:
Appendix B discusses architectures such as ResNets, modified MLPs, and SPINNs conceptually, but there is no direct experimental comparison to SPINN itself, and PirateNet—arguably the strongest empirical baseline—is not meaningfully incorporated into the mechanism discussion. As a result, the conceptual positioning of the paper against nearby alternatives feels incomplete.

-   The role of LM is not fully disentangled from the role of the architecture:
The paper explicitly states that LM can, in principle, be applied to general PINN parameterizations, but is practical mainly for small models because its cost scales with the number of trainable parameters. This makes it somewhat unclear how much of the gain comes from the proposed time-induced parameterization itself versus from operating in a compact regime where LM is tractable.

-   The precision mismatch across methods is a real fairness concern:
In the main results, all non-TINN baselines are trained in single precision, whereas TINN is trained in double precision because LM requires solving linear systems stably. Since numerical precision can materially affect optimization stability and final error, this makes the comparison somewhat harder to interpret cleanly.

-   The Adam ablation is helpful but not fully convincing:
While Table 3 suggests that the gains are not purely due to LM, the improvements over PirateNet are modest on Burgers and Klein–Gordon, and the comparison also uses different iteration counts in the appendix settings. For that reason, this ablation supports the paper’s story, but does not fully establish that the gains are primarily architectural.

---

> ### Author Rebuttal · Authors · 2026-03-31
>
> We thanks for the constructive feedback. In the revision, we will incorporate **all** of the reviewer's feedback and clarify the relevant points.
> ### W1 + W2: Theory
> Prop. 5.1 is mainly a representation result showing that classical universal approximation still applies, so TINN does not lose expressivity. Our main contribution is instead a simple architectural prior: layer-wise temporal embedding, which injects time across depth and improves optimization through structured modulation of spatial features. The discussion in “TINNs Provide Flexibility...” is also viewed as an optimality comparison. A standard MLP must realize the full solution with one static parameter set under the constraints in L425-429, whereas TINN is more flexible. A sharper theory is beyond the scope, as it would require joint analysis of optimization dynamics and architecture. Yet, consistent gains across nonlinear and higher-dimensional PDEs support the practical value of this inductive bias.
> ### W3: Positioning and comparisons
> Our goal is not to promote a specific backbone, but to present TINN as a plug-and-play structural prior. We use MLPs in the main experiments to isolate the effect of temporal modulation. We also add a direct comparison with SPINN on the (3+1)-D Klein-Gordon (KG) eq., where TINN achieves 2.36E-06 versus 9.3E-03. PirateNet is a strong complementary baseline, improving architecture and causality-aware training, whereas TINN introduces structured temporal modulation.
> ### W4 + Q3 + W6: Disentangling architecture, optimizer, and fairness
> To disentangle the effects of LM, model size, and training setup, we run matched experiments with TINN, MLP, and PirateNet under the same LM optimizer and similar parameter counts (~1K). TINN outperforms both baselines across all benchmarks, indicating that the gains are not due to an LM-friendly regime (see table).
>
> | | **Burgers** | **Allen–Cahn** | **KG** |
> |---|---|---|---|
> | TINN | **6.89E-07** | **3.85E-06** | **4.78E-06** |
> | MLP | 1.92E-05 | 7.14E-06 | 5.11E-06 |
> | PirateNet | 1.94E-06 | 4.09E-05 | 8.16E-01 |
>
> Second, we perform an Adam-based ablation with matched budgets (~530K parameters, 100K iterations), where TINN is comparable on Burgers (3.25E-05 vs. 2.18E-05) with lower time (0.85hr vs. 1.16hr), and significantly better on Allen–Cahn (1.36E-04 vs. 1.64E-03). Together with the original ablations (TINN vs. PirateNet w/ Adam and TINN vs. MLP w/ LM), these results consistently show that the improvements stem from the proposed time-induced parameterization rather than optimizer choice, model size, or training setup.
> ### W5: Precision
> TINN uses double precision (DP) mainly as LM requires solving linear systems stably. A fairer test is whether the advantage remains when strong baselines also use DP. On Burgers, DP gives only modest or inconsistent gains for CoPINN* and PirateNet, while incurring substantial overhead, and does not close the gap to TINN. This suggests the main advantage comes from TINN’s parameterization rather than precision alone.
> | | **SP** | **DP** |
> |---|---|---|
> | CoPINN* | 1.03E-04 / 0.78hr | 2.72E-05 / 5.92hr |
> | PirateNet | **1.46E-06** / 1.70hr | **3.14E-06** / 16.8hr |
> ### Q1: Choice of PDEs and baselines
> We choose CoPINN and PirateNet as strong, complementary baselines for explicit space-time decomposition and causality-aware optimization. The benchmarks are standard time-dependent PDEs covering shocks, phase transitions, nonlinear wave propagation, and oscillatory dynamics, making them suitable for testing TINN’s ability to reduce time entanglement. We also include a harder chaotic case, where TINN achieves 8.58E-03 vs. 3.86E-02 for PirateNet on Kuramoto-Sivashinsky.
> ### Q2: On $\partial_x u(x,t)$.
> When gradients sharpen over time, as in Burgers’ eq., the solution develops time-dependent high-frequency spatial structure. A standard space-time MLP must represent all such variation in one shared feature space across time. TINN instead uses layer-wise temporal modulation, which acts as a time-dependent reparameterization of the spatial basis and reduces this burden.
> ### Q4: Explanation of time-embedding
> By *macro-level coherence*, we mean that layers evolve in a temporally synchronized way; by *micro-level diversity*, we mean that the magnitude of this variation differs across layers. To verify these effects, we solve the KdV eq. by time stepping and fit, at each step, a two-hidden-layer MLP in $x$. The resulting parameter trajectories $\{W_\ell(t), b_\ell(t)\}$ clearly exhibit both patterns. Moreover, the evolution is strongly low-rank: the top singular component explains 93%-99% of the weight variation and 87%-89% of the bias variation.
> ### Q5: Initial and periodic BCs
> The issue comes from incompatibility with periodicity: although $u(x,0)=x^2\cos(\pi x)$ matches values at $x=\pm1$, its spatial derivatives do not. Hence the initial condition is not genuinely periodic and induces a residual near the initial time that training cannot eliminate.

---

> > ### Author Rebuttal · Reviewer_EJzp · 2026-04-03
> >
> > Thank you for the detailed rebuttal. The additional experiments and clarifications are helpful. In particular, the added LM-controlled comparisons, the direct comparison against SPINN, and the clarification on the incompatibility between the initial condition and periodic boundary conditions address part of my experimental concerns. I also appreciate the additional discussion regarding optimizer choice and precision.
> >
> > However, I am keeping my score unchanged because my main concern remains the explanatory and theoretical support of the paper. For acceptance at ICML, I do not yet find the current level of justification sufficient. In particular, the responses to Q2 and Q4 remain mostly intuitive rather than analytic. The discussion of why a standard space-time parameterization may struggle when spatial gradients sharpen over time still reads more like an early-stage articulation of the core idea than a sharp mathematical or mechanistic explanation. Likewise, the explanation of the layer-wise time embedding remains too abstract; I still do not have a concrete understanding of how the temporal embedding changes the layer parameters over time and why this should lead to the claimed optimization benefits.
> >
> > Relatedly, the paragraph titled “TINNs Provide Flexibility ...” still reads more like an early-stage architectural intuition than a sufficiently developed explanation of when and why TINN should outperform nearby alternatives. Since the empirical scope, while improved, is still somewhat limited across problem classes and architectures, I believe this explanatory gap remains important.
> >
> > Overall, I think the rebuttal partially resolves some experimental concerns, but the remaining issues concern the core justification and interpretation of the method, and these are not easy to address within a short rebuttal.

---

> > > ### Author Response · Authors · 2026-04-05
> > >
> > > We thank the reviewer for the thoughtful follow-up. Below we give a more concrete mechanistic explanation for Q2/Q4.
> > >
> > > **Q2: TINN helps when spatial gradients sharpen over time.**
> > >
> > > Consider the one-hidden-layer space-time model
> > > $$
> > > u\_M(x,t)=\sum\_{j=1}^m c\_j\sigma(w\_jx+v\_jt+b\_j).
> > > $$
> > > If $\sigma$ has bounded derivatives (e.g., $\sigma=\tanh$), then
> > > $$
> > > \|\partial\_x u\_M(\cdot,t)\|\_\infty \le \|\sigma'\|\_\infty \sum\_{j=1}^m |c\_jw\_j|,
> > > \qquad
> > > \|\partial\_{xx}u\_M(\cdot,t)\|\_\infty \le \|\sigma''\|\_\infty \sum\_{j=1}^m |c\_j||w\_j|^2,
> > > $$
> > > and
> > > $$
> > > \|\partial\_{xt}u\_M(\cdot,t)\|\_\infty \le \|\sigma''\|\_\infty \sum\_{j=1}^m |c\_j||w\_jv\_j|,
> > > $$
> > > These bounds are independent of $t$, so spatial and mixed derivatives are controlled by a single time-independent envelope.
> > >
> > > When the target evolves from smooth to sharp (e.g., Burgers), a single parameter set must capture both regimes. Large $w\_j$ enables steep gradients but introduces high-frequency artifacts at early times; small $w\_j$ preserves smoothness but limits late-time resolution. This intrinsic trade-off reflects time-entanglement between spatial scale and temporal evolution.
> > >
> > > For TINN
> > >
> > > $$
> > > u\_T(x,t)=\sum\_{j=1}^m c\_j(t)\sigma(w\_j(t)x+b\_j(t)),
> > > $$
> > > the analogous bounds become
> > > $$
> > > \|\partial\_x u\_T(\cdot,t)\|\_\infty \le \|\sigma'\|\_\infty \sum\_{j=1}^m |c\_j(t)w\_j(t)|,
> > > \qquad
> > > \|\partial\_{xx}u\_T(\cdot,t)\|\_\infty \le \|\sigma''\|\_\infty \sum\_{j=1}^m |c\_j(t)||w\_j(t)|^2,
> > > $$
> > > so the envelope varies with $t$. This enables adaptive spatial scaling over time and removes the above coupling.
> > >
> > > **Q4: Layer-wise time embedding.**
> > >
> > > We do not use an arbitrary hypernetwork for $\theta(t)$. Instead, we introduce a low-dimensional temporal code
> > > $$
> > > \Phi(t)=(1-\alpha)t+\alpha\odot N(t),
> > > \qquad
> > > \Phi(t)\in\mathbb{R}^{2L}
> > > $$
> > > (or $\mathbb{R}^{2L-1}$ when the output bias is omitted), and lifts it to the full parameter vector through an entrywise affine map. For parameter group $i$ and entry $p$,
> > > $$
> > > \theta\_{i,p}(t)=a\_{i,p}\Phi\_i(t)+b\_{i,p}.
> > > $$
> > >
> > > *Concrete example.* For a one-hidden-layer network, this yields
> > > $$
> > > u(x, t) = \sum^m\_{j=1}(\Phi\_3(t)c\_j+\tilde{c}\_j) \sigma\left((\Phi\_1(t)w\_j + \tilde{w}\_j)x + (\Phi\_2(t)b\_j + \tilde{b}\_j)\right).
> > > $$
> > > Here, $\Phi\_1(t)$ rescales the input weights, controlling spatial frequency, while $\Phi\_3(t)$ modulates the output amplitude. If $\Phi\_1(t)$ increases over time, the network progressively sharpens its spatial features.
> > >
> > > This makes the two effects precise.
> > >
> > > *Macro-level coherence.* All entries in the same group share the same scalar driver $\Phi\_i(t)$. They may differ in magnitude and sign through $a\_{i,p}$, but they follow the same temporal profile. More globally, all groups are driven by coordinates of the same low-dimensional code $\Phi(t)$. Thus the network evolves on a low-dimensional temporal manifold, rather than assigning an independent time function to each parameter.
> > >
> > > *Micro-level diversity.* Despite the shared driver within a group, different entries still have different affine coefficients $a\_{i,p},b\_{i,p}$. Hence they need not have the same scale, sign, or offset, and different groups may follow different coordinates $\Phi\_i(t)$. The model therefore enforces shared temporal structure without forcing identical parameter motion.
> > >
> > > This intuition can be formalized as a low-dimensional trajectory. For each group,
> > > $$
> > > \bar\theta\_i(t)=a\_i\Phi\_i(t)+b\_i,
> > > $$
> > > so its trajectory lies in a two-dimensional affine subspace, and after centering in a one-dimensional linear subspace. Thus, each group evolves along a single dominant temporal mode.
> > >
> > >  This is consistent with a diagnostic experiment on the KdV equation, where we fit an independent spatial network at each time step using a classical solver and observe that the resulting parameter trajectories are nearly rank-1: the leading singular component explains 93% to 99% of the variation for weights and 87% to 89% for biases.
> > >
> > > Our embedding is therefore not an abstract heuristic; it is a compact parameterization designed to match the observed low-dimensional evolution of the fitted per-time networks.
> > >
> > > In summary, a full convergence theory would require a joint analysis of architecture, optimizer, and training dynamics, which is beyond the scope of the present work. Our point here is narrower: the characterization above gives a concrete mechanistic foundation in terms of spectral adaptation and low-rank parameter evolution. If the reviewer has a specific type of theorem in mind, we would be grateful to understand that direction and consider it as an important avenue for future work.

---

### Official Review · Reviewer_TVFT · 2026-03-13

**Soundness:** 2
**Presentation:** 2
**Significance:** 2
**Originality:** 2
**Overall Recommendation:** 2
**Confidence:** 4

**Summary:**

This paper argues that, for time-dependent PDEs, conventional PINNs based on spatial-temporal coordinates suffer from a time-entanglement issue, because both spatial and temporal information are processed by a single shared network and different temporal regimes must therefore be encoded within the same representation. To address this, the authors propose TINN, where time is no longer fed directly as an input; instead, the backbone parameters vary as a function of time. Rather than using a heavy full hypernetwork, the method employs a lightweight layer-wise temporal code together with an efficient parameter generation scheme, and is trained with an LM-based optimization strategy.

**Compliance With Llm Reviewing Policy:**

Affirmed.

**Key Questions For Authors:**

1. Since the core claim of the paper is closely related to explicit space-time separation methods and modulated-INR-style approaches, could the authors provide direct comparisons against such baselines?

2. Would TINN remain effective on higher-dimensional PDEs, such as 3D Navier-Stokes?

3. The layer-wise shared temporal code is an interesting design choice, but it would be helpful to understand more clearly what kinds of temporal dynamics it can capture well, and under what settings it may fail.

**Limitations:**

See the weaknesses and questions above.

**Strengths And Weaknesses:**

**Strengths**
1. The proposed methodology is clear. The paper explains the limitation of standard PINNs as a structural representational bottleneck and introduces TINN as a principled remedy.

2. The method shows strong empirical performance across the PDE benchmarks considered in the paper.

**Weaknesses**
1. The experimental scope is somewhat limited. Important settings such as more complex geometries, higher-dimensional problems, and inverse problems are not included, so it is still unclear whether the method generalizes to more realistic scientific computing scenarios.

2. The effects of the architecture and the optimizer are not fully disentangled. The reported gains appear to rely substantially on the TINN + LM combination, and a more direct analysis would strengthen the paper.

3. Although the paper discusses related directions such as Causal PINNs and explicit space-time separation methods, these are not included in the main empirical comparison.

4. The discussion of hypernetworks is limited to complexity considerations, and there is no direct empirical comparison against hypernetwork-based baselines.

---

> ### Author Rebuttal · Authors · 2026-03-31
>
> # Response to Reviewer TVFT
> We thank Reviewer TVFT for the constructive feedback and the opportunity to clarify our work. We address the specific concerns below.
> ### W1 + Q2: Regarding the experimental scope and the effectiveness of TINN in high-dimensional problems.
> We thank the reviewer for highlighting the importance of broader experimental coverage. TINN scales naturally to higher-dimensional PDEs: although the input dimension increases, the hypernetwork output remains independent of spatial dimensionality, avoiding parameter blow-up. In a new rebuttal experiment on the $(3+1)$-D Klein–Gordon equation, TINN achieves a relative $L^2$ error of $2.36 \times 10^{-6}$ versus $9.3 \times 10^{-3}$ for SPINN, using only $1225$ parameters. For more complex settings (e.g., irregular geometries and inverse problems), TINN is orthogonal and can be seamlessly combined with existing PINN extensions. In this sense, TINN serves as a **plug-and-play structural prior** across architectures. We will include additional experiments in the camera-ready version.
> ### W2: Regarding the disentanglement of TINN and the LM optimizer.
> We agree that TINN combined with LM yields strong performance and appreciate the request for clearer disentanglement. In the original submission, we compare (i) TINN vs. PirateNet under Adam (Table 3), isolating architectural effects, and (ii) TINN vs. MLP under LM (Table 4), evaluating architecture under a fixed optimizer. To further strengthen this, we include a new rebuttal experiment where MLP, PirateNet, and TINN are all trained with LM; despite similar parameter counts, TINN consistently achieves the lowest error across all benchmarks. These results show that the gains arise from the architectural design, not solely from the use of LM.
> | Case | Rel $L^2$-error | Parameters |
> | :--- | :--- | :--- |
> | **Burgers** | | |
> | TINN | **6.89E-07** | 1145 |
> | MLP | 1.92E-05 | 1160 |
> | PirateNet | 1.94E-06 | 1190 |
> | **Allen-Cahn** | | |
> | TINN | **3.85E-06** | 1185 |
> | MLP | 7.14E-06 | 1202 |
> | PirateNet | 4.09E-05 | 1190 |
> | **Klein-Gordon** | | |
> | TINN | **4.78E-06** | 1185 |
> | MLP | 5.11E-06 | 1202 |
> | PirateNet | 8.16E-01 | 1190 |
> ### W3 + Q1: Regarding Causal PINN and explicit space-time separation.
> We thank the reviewer for emphasizing comparisons with causality-aware and space–time separation methods. We use PirateNet as a strong causality-aware baseline and note that TINN’s layer-wise modulation is orthogonal and can be combined with such approaches. Compared to explicit separation methods, TINN uses implicit layer-wise modulation via a hypernetwork. In a new rebuttal experiment, TINN achieves $8.42 \times 10^{-10}$ vs. $3.04 \times 10^{-8}$ on a transport equation. We will include further comparisons in the camera-ready version.
> | Case | Rel $L^2$-error |
> | :--- | :--- |
> | **Transport Equation** | |
> | TINN | **8.42E-10** |
> | explicit Separation Method (swim) | 3.04E-08 |
> ### W4: Regarding the hypernetwork baselines.
> To compare with hypernetwork baselines, we include a new rebuttal experiment against a HyperPINN-style method (Belbute-Peres et al., 2021) with matched parameter budgets. While HyperPINN globally generates all backbone parameters, TINN uses layer-wise temporal embeddings, providing a more structured and efficient modulation. Empirically, TINN outperforms HyperPINN on Burgers and Klein-Gordon and remains competitive on Allen-Cahn, while requiring less training time, supporting the effectiveness of layer-wise modulation.
> | Method | Rel $L^2$-error | Time | Parameters |
> | :--- | :--- | :--- | :--- |
> | **Burgers** | | | |
> | HyperPINN | $1.94\text{E-}06 \pm 9.34\text{E-}07$ | 0.90hr | 1162 |
> | TINN | **6.89E-07 $\pm$ 3.97E-07** | 0.75hr | 1145 |
> | **Allen-Cahn** | | | |
> | HyperPINN | **3.49E-06 $\pm$ 1.71E-06** | 0.83hr | 1202 |
> | TINN | $3.85\text{E-}06 \pm 1.48\text{E-}06$ | 0.78hr | 1185 |
> | **Klein-Gordon** | | | |
> | HyperPINN | $2.02\text{E-}01 \pm 3.00\text{E-}01$ | 0.77hr | 1202 |
> | TINN | **4.78E-06 $\pm$ 2.63E-06** | 0.67hr | 1185 |
> ### Q3: Regarding layer-wise shared temporal code.
> The layer-wise temporal embedding in TINN modulates intermediate representations across depth, allowing the network to adapt its spatial basis over time without explicit space–time factorization. This enables it to capture evolving structures, such as the transition from smooth to shock-like profiles in Burgers’ equation. We further evaluate this on the Kuramoto-Sivashinsky equation, where TINN achieves a relative $L^2$ error of $8.58 \times 10^{-3}$, improving over $3.86 \times 10^{-2}$ for PirateNet, demonstrating its ability to capture complex temporal dynamics. As a limitation, the modulation can be viewed as a structured (low-rank) parameterization, which may be less expressive for highly multi-scale temporal features. Nevertheless, it provides a favorable trade-off between expressivity and efficiency, enabling stable training with a compact parameter budget.

---

> > ### Author Rebuttal · Reviewer_TVFT · 2026-04-04
> >
> > Thank you for the rebuttal. However, I do not believe that the rebuttal has sufficiently addressed the main concerns raised in my review. In particular, I remain concerned that the empirical comparison against relevant baselines is still not sufficiently comprehensive and that the experimental scope remains limited. Therefore, I will keep my current score.

---

> > > ### Author Response · Authors · 2026-04-07
> > >
> > > We thank the reviewer for the follow-up comment. In the rebuttal, we directly addressed the main concerns raised in the original review by adding new experiments on: (i) **higher-dimensional PDEs**, (ii) **disentangling the effect of the architecture from that of the LM optimizer** under matched parameter budgets, (iii) **comparison against an explicit space-time separation baseline**, and (iv) **comparison against a hypernetwork-based baseline**. Across all of these settings, TINNs consistently outperformed the alternatives. These additions were introduced specifically to address the concerns raised in the reviewer’s original comments.
> > >
> > > To further address the concern that the empirical study is "still not sufficiently comprehensive" and that the scope "remains limited," beyond the additions already included in our first-round response, we further evaluate TINN on higher-dimensional settings (Navier–Stokes and flow mixing), more challenging PDE dynamics (flow mixing and Kuramoto–Sivashinsky eq.), and inverse problems (Burgers' eq.). Baseline results are taken directly from the corresponding original papers.
> > >
> > > *   **(3+1)D Navier–Stokes eq.**
> > >     We further test TINN on the same example considered in SPINN. On this benchmark, TINN achieves a vorticity relative $L^2$ error of *4.05E-04*, outperforming the 1.9E-03 reported in SPINN.
> > >
> > > *   **(2+1)D Flow Mixing Problem.**
> > >     We also evaluate on the flow-mixing benchmark from SPINN. Although the governing equation is linear, the problem remains challenging because the spatially varying rotational field induces strong shear, leading to rapid spectral enrichment, increasingly oscillatory solution structure, and pronounced multi-scale behavior. These effects manifest as thin, dynamically evolving filaments with large gradients, which are difficult for standard PINNs to resolve. In addition, the PDE is non-separable, so the space-time variables are strongly coupled rather than well approximated by a simple separated representation. Under the same training budget, TINN attains a relative $L^2$ error of *9.45E-04*, outperforming the 2.9E-03 reported for SPINN.
> > >
> > > *   **Chaotic Kuramoto–Sivashinsky eq.**
> > >     We additionally evaluate TINN on the Kuramoto–Sivashinsky eq (see also the response to Reviewer EJzp), which exhibits nonlinear instability and spatiotemporal chaos. Using ten time windows, TINN obtains a relative $L^2$ error of *8.58E-03*, outperforming the 3.86E-02 achieved by PirateNet.
> > >
> > > *   **Inverse problem for Burgers' eq.**
> > >     We consider
> > >     $$u_t + \lambda_1 u u_x - \lambda_2 u_{xx} = 0,$$
> > >     and follow the same setup as in PINN, where $\lambda_1 = 1$, $\lambda_2 = 0.01/\pi$, and 2000 observations are provided. TINN recovers the coefficients with absolute errors *9.25E-09* for $\lambda_1$ and *7.72E-09* for $\lambda_2$, substantially outperforming PINN, which reports 9.6E-04 and 4.69E-03, respectively.
> > >
> > > Taken together, the experiments in our first-round response and the new results provided here comprehensively demonstrate that TINN consistently outperforms the baselines across diverse challenging settings.
> > >
> > > At the same time, we understand that the reviewer may wish to see a broader empirical study. Because the second-round feedback does not identify specific additional baselines or experiments, it is difficult for us to determine the most appropriate further additions within the limited rebuttal period, especially since the additional experiments were designed to directly address the reviewer’s original concerns. While a rebuttal cannot reasonably cover all possible extensions, we hope the paper will be assessed based on whether the main concerns raised in the review have been substantively addressed by the new evidence provided.

---

### Decision · Program_Chairs · 2026-04-30

**Decision:**

Accept (regular)

**Comment:**

This paper received a mixed reviews in the beginning. The main concerns reviewers raised include:

- **Limited empirical results.** Reviewers (particularly TVFT) requested additional experiments across different setups, direct comparisons to disentangle the architectural effects of TINNs from the LM optimizer (both TVFT and EJzp), and tests of the method's scalability (Ydz2).
- Limited theoretical contributions (EJzp).
- Discussion and position with related works are not sufficient (EJzp, Ydz2)
- Concerns were raised regarding the consistency of the experimental setups, precision mismatches, and metrics across different baselines (EJzp, KAeR, Ydz2).

During the rebuttal, the authors provide extensive additional experiment results. These results have addressed reviewer Ejzp, Ydz2, and KAeR’s concerns regarding experiments. While TVFT insists in negative score, citing that the comparison with relevant baselines are not comprehensive. However, TVFT didn’t provide which specific experiment or baseline were missing after the rebuttal. After reading the results provided by the reviewers and the rebuttal, I am aligned with the majority reviewer’s judged and would like to recommend the paper for acceptance.